**Spatial Agents for Geological Surface Modelling**
Eric A. de Kemp[1]
Correspondence to: Eric A. de Kemp (edekemp@canada.ca)
[1]Geological Survey of Canada,
Three-dimensional Earth Imaging and Modelling Lab
601 Booth Street, Ottawa, Canada, K0E 1E9
E-mail:  eric.dekemp@canada.ca
https://orcid.org/0000-0003-0347-5792
Tel.: 01-613-867-8812
**Abstract**
Increased availability and use of 3D rendered geological models has provided society with predictive capabilities, supporting
natural resource assessments, hazards awareness and infrastructure development. The Geological Survey of Canada, along
with other such institutions, have been trying to standardize and operationalize this modelling practice.  Knowing what is in
the subsurface, however is not an easy exercise, especially when it is difficult or impossible to sample at greater depths.
Existing approaches to creating 3D geological models involves development of surface components that represent spatial
geological features, horizons, faults and folds, and then assembling them into a framework model as context for down-stream
property modelling applications (geophysical inversions, thermo-mechanical simulations, fracture density models etc.).  The
current challenge is to develop reasonable starting framework geological models from sparser data regions, when we have
more complicated geology. This study explores this problem of geological data sparsity and presents a new approach that
may be useful to open up the log jam in modelling the more challenging terrains using an agent-based approach.
Semi-autonomous software entities called spatial agents can be programmed to perform spatial and property interrogation
functions, estimations and construction operations for simple graphical objects, that may be usable in building three-
dimensional geological surfaces. These surfaces form the building blocks from which full geological and topological models
are built and may be useful in sparse data environments, where ancillary or a-priori information is available. Critical in
developing natural domain models is the use of gradient information. Increasing the density of spatial gradient information
(fabric dips, fold plunges, local or regional trends) from geologic feature orientations (planar and linear) is key to more
accurate geologic modelling, and core to the functions of spatial agents presented herein. This study, for the first time,
examines the potential use of spatial agents to increase gradient constraints in the context of the Loop project
(https://loop3d.github.io/) in which new complementary methods are being developed for modelling complex geology for
regional applications. The Spatial Agent codes presented may act to densify and supplement gradient, and on-contact control
points, used in *LoopStructural*  (www.github.com/Loop3d/LoopStructural) and *Map2Loop*
*(*https://doi.org/10.5281/zenodo.4288476).
Spatial agents are used to represent common geological data constraints such as interface locations and gradient geometry,
and simple but topologically consistent triangulated meshes. Spatial agents can potentially be used to develop surfaces that
conform to reasonable geological patterns of interest, provided they are embedded with behaviors that are reflective of the
knowledge of their geological environment. Initially this would involve detecting simple geological constraints; locations,
trajectories and trends of geological interfaces.  Local and global eigenvectors enable spatial continuity estimates, which can
reflect geological trends, with rotational bias, using a quaternion implementation. Spatial interpolation of structural geology
orientation data with spatial agents employs a range of simple nearest neighbour to inverse distance weighted (IDW) and
quaternion based spherical linear interpolation (SLERP) schemes.  This simulation environment implemented in NetLogo 3D
is potentially useful for complex geology - sparse data environments where extension, projection and propagation functions
are needed to create more realistic geological forms.
Keywords – spatial agents, three-dimensional geological model, simulation, surfaces
**1        Introduction**
The major challenge that this paper is trying to address is the breakdown in achieving geologically realistic model results
from sparse data in more complicated geological scenarios when using the existing methods and algorithms. This is no doubt
a problem in other modelling domains as well, but is acute in geological applications, where access to data in the subsurface
is often extremely expensive, terrain access prohibitive, or the depth of investigation too extreme for direct sampling and
must rely on coarser geophysical methods that often do not adequately image the features being modelled.  This paper
explorers the use of extension, propagation and cohesion methods, which can be considered part of 'swarm' technology,
using spatial agents in an attempt to deal with this challenge.
Geological modelling covers a wide range of applications and domains from thermo-mechanical modelling (Cloetingh et al.,
2013) to basin analysis (Barrett et al., 2018), mineral potential estimation (Skirrow et al., 2019) in 3D (Hu et al., 2020;
Sprague et al., 2006) and even 4D applications (Parquer et al., 2020; White, 2013). Herein we focus on the starting
framework model, the stratigraphic and structural surface model that provides the initial context for these more down-stream
property embedded modelling efforts.  Generally, these geological models can be represented as BREP (Boundary
Representation) models (Pellerin et al., 2017; Caumon et al. 2009) but recently many of these are defined through implicit
derived surfaces with topologically encoded volumes (Grose et al., 2021; de la Varga et al., 2019; Wellmann et al., 2019;
Grose et al., 2017; Laurent et al. 2016; Hillier et al. 2014, Frank et al. 2007; Courrioux et al., 2001; Lajaunie et al., 1997). In
each case the accuracy of the BREP and/or implicit surface model features such as horizons, folds or faults, are dependent on
the quality of the geological input data that is available, but also importantly, on the algorithms and methods used to build
them (Wellmann and Caumon, 2018; MacCormack and Eyles, 2012).
Existing methods applied to the combined sparse data and complex geology scenario, will tend to produce holes, gaps and
feature drop-outs, away from control data, as well as arbitrary horizon thickness changes that combine to give a geologically
unreasonable bubble gum look to these models (Fig. 1). Current methods in sparse data configurations tend to bias for these
unrealistic geometries using radial based kernel functions, optimized for local smoothness in order to achieve a mathematical
solution (Hillier et al., 2021; Hillier et al., 2014). This often comes at the price of geological realism (Hillier et al., 2021;
MacCormack and Eyles; 2012). Is it possible that, with a new approach, geological features could be more realistically
modelled by using spatial agents to 'fill-the-gaps' in the process?
Section 1 provides an overview, context and review for the current study, surveying various application domains with an eye
toward natural and more specific earth sciences agent applications. Section 2 outlines the use of spatial agents for structural
geology. A summary of current geological surface modelling approaches is given, with some argumentation that highlights
the need for new approaches particularly when data is sparse, and geology is more complex. The mechanisms for using
constraints, inter-agent communication and characterization of required behaviors. A summary is given of the critical
intrinsic properties of spatial agents that may aid in future research in this area. In section 3 several spatial agent demos are
used to represent simple contact surfaces as agent constructed triangular meshes, fold closures and simulations of unmeshed
structural swarms from sparse points. There are 6 main programs, each highlighting critical functionality that will be required
should structural agents be developed into a more complete geomodelling system in the future. Lastly, section 4 provides a
discussion for how structural agents could be applied and some final conclusions from the study.

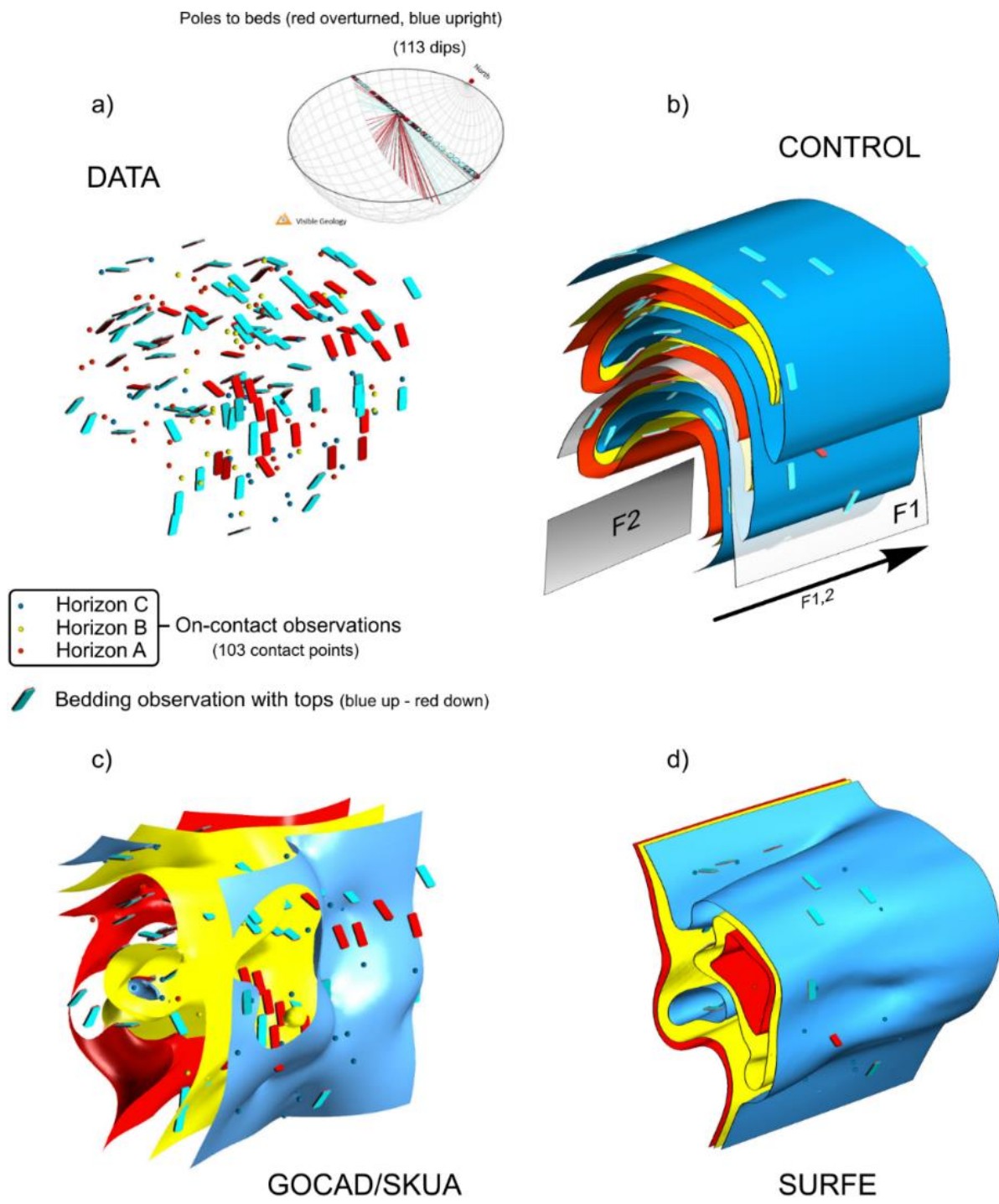

**Figure 1.** Comparison of synthetic geological three-dimensional models of classic Ramsay type 3 interference folds (Ramsay, 1967;1962), with identical

data. (a) Uniaxial dip data, with local opposing tops, represented on equal-angle Wulf plot (https://app.visiblegeology.com/stereonet.html).     (b) Control

model developed with SPARSE (de Kemp et al., 2004), with F1-F2 horizontal, north trending hinges, (c) implicit surface models with Gocad/SKUA (Jayr et

al., 2008) and (d) SURFE (Hillier et al., 2104).

## 1.1 Agent Challenge

Spatial Agents are virtual spatial entities that have freedom to interact with each other and their environment, which can include various domain data, in order to solve a well-defined problem, for example to predict the growth of an urban centre, an ant hill or the course of a meandering river system under variable rain fall and soil conditions. Some of the core characteristics of spatial agents could potentially be used to essentially 'grow' features away from the control data, keeping them intact while extending and respecting regional gradient information. In a sense similar to how the human mind might fill-in through geological interpretation of a map or cross-section.

The Loop effort is attempting to address this ongoing challenge (Ailleres et al., 2019) that tends to present itself when geology becomes more complicated, with more elaborate geo-histories, for example, geo-histories with early cryptic sedimentary and volcanic depositional cycles, and a spectrum of brittle to deeper crustal deformation events, and through masking metamorphic processes. Geo-histories with overprinting intrusive events, from thin dyke swarms to consuming batholithic intrusions can also completely erase all macroscopic evidence of earlier processes. The challenge is most acute when the data required to accurately model these scenarios is quite limited. It is in these in-land frontier zones, where most of our data is only at ground surface, interpreted from remote sensing images, or sparingly at depth, with clustered spatially biased drill holes near mineralized zones. These regions may have been surveyed with geophysical instruments, and the data used to derive models representing at depth rock property distributions for density and magnetic susceptibility, conductivity and resistivity. However, in almost all cases there is a lack of high-resolution geophysics, as 2D or 3D seismic data, from these surveys, which is more commonly available and used in the practice of hydrocarbon reservoir modelling workflows. The suggestion, presented in this study, is that we may be able to better face some of the sparse data conditions, characteristic of more complex geological terrains, by taking advantage of the properties that spatial agents posses. Primarily for spatial agents to densify input constraints for horizon dips, better model the local structural trends or anisotropy, and extend features such as regional fold plunges. These derived constraints could be useful as supplemental input to *LoopStructural* (Grose et al., 2021) and *Map2Loop* (Jessell et al., 2021) to increase the accuracy and geological reasonableness of those downstream models.

This study highlights the potential use of Spatial Agents in the context of the Loop project (Ailleres et al., 2019) that is developing new methods supporting the modelling of more complex geological terrains. With this initial study, which is a

first to highlight their potential use for sparsely constrained complex geology, we may inspire more development in this area
and complement the various new methods that emerge from Loop, and hopefully other initiatives in the future.
**1.2     Agent Applications**
In general, an agent-based system is used to see the effects of autonomous individuals, groups or objects on the overall system
when solutions are onerous and/or computationally expensive. A global algorithm involving a single large multi-parameter
matrix inversion may take many days to compute with a single outcome, but an agent-based model may be able to produce
several outcomes in minutes or hours (Siegfried, 2014). Agent-based models have their roots in the development of cellular
automata and complexity theory, which has been able to model complex natural and artificial systems with simple
neighbourhood algorithms (Cervelle and Formenti, 2009; Wolfram, 1994; Von Neumann, 1966). Agent applications are
extensively used in the entertainment industry (Damiano et al., 2013); computer games for sports and battle simulation (Zuparic
et al., 2017; Guo and Sprague, 2016), landscape and land use design, management and visualization (Tieskens et al., 2017;
Valbuena et al., 2010); urban planning (Motieyan and Mesgari, 2018; Levy et al., 2016); crowd modelling for public transport
and community infrastructure design (Dickinson et al., 2019; Hoy and Shalaby, 2016); climate change and adaptation modelling
(Amadou et al., 2018); Architecture and Engineering design (Guo and Li, 2017; Van Dyke Parunak et al., 2001) as well as
hazard response and real-time three-dimensional mapping (Schlögl et al., 2019; Bürkle 2009); transportation and surveillance
using semi-automated or fully-autonomous vehicles such as drones and automobiles (Fagnant and Kockelman, 2014; de Swarte
et al., 2019). Agent-based modelling has been used in the Earth Sciences for spatial-temporal more process-oriented modelling
such as solar storm and flare activity (Schatten, 2013), Groundwater modelling (Jaxa-Rozen et al., 2019) and Earthquake
prediction (Azam et al., 2015) to name a few examples.
These applications generally do not use trend information, or what structural geologists refer to as anisotropy, and gradient type
information such as horizon dip data, with polarity, or direction, which the structural agents do in this study, however these
diverse applications do have some common elements that software agents are well suited to. The problem domains have multi-
scalar environments; molecular to planet scale, with local or global model element interactions, and non-linear, multi-source
physical dependencies. Agents could be interacting at molecular scale with quantum-mechanical, ionic and thermodynamic
influences, for example, for protein-folding (Semenchenko et al., 2016; Nelson et al., 2000), for a visual demonstration of
molecular agent simulation see: https://www.youtube.com/watch?v=4Z4KwuUfh0A or at galactic scale
http://www.gravitysim.net/index.html. The ability to operate in a non-centralized control structure, being sensitive to other
neighbours conditions and geometric states as well as their ability to respond to local or globally changing conditions may give
spatial agents an advantage. Their independence allows them to operate as individual elements, for example a single point
observation, or to work collectively as a team or 'swarm'. This allows the application of agent rules that may determine local
cohesion levels and shape characteristics as well as changes of state depending on specific conditions such as moving in a
direction, stopping, or spawning other processes. This allows them to behave in a flexible and efficient manner, without the need
for global partitioned data structures or tightly coupled deterministic algorithms.  Many agent examples are biologically based
such as the classic flock of birds examples; 'murmurings' and geese in V-formation, beehive  and anthill construction examples
(Mnasri et al., 2019; Carrillo et al., 2014; Johnson and Hoe, 2013). These examples highlight the potential to capture multi-scaler
and complex interaction that has enhanced the uptake of this technology for medical and biology fields (An et al., 2017; Rigotti
and Wallace, 2015).
**1.3      Agent Characteristics**
Agents operate as semi-autonomous software entities that are not directly controlled by any centralized command structure
and can operate with a great deal of independence from each other. They are programed with roles, beliefs and behaviors that
can be triggered by the state of their local or regional environment. They can interact with other similar or different agents to
collectively achieve a goal, acting like a swarm.  For example, considering a construction simulation game, a carpenter would
be considered a single agent that could be assigned the framing role to construct a house. The house in this case would be an
example of a single Agent-based Model (ABM). If there are many agents with different tasks but working collectively,
perhaps a team of framers with a foreman, an architect and a designer, working on a larger more complex building, this
would be a Multi-ABM (MABM).  When two, three or four dimensional maps or entities with spatial properties critical in the
modelling process are involved, this is characteristic of Spatial Agent-Based Models (SABM). Spatial agents and spatial
multi-agent-based modelling systems (SABS and SMABS), or the non-spatial agent-based models (ABM) form a family of
approaches which have been used in a wide range of applications that take advantage of the efficiencies and freedoms that
these systems possess (Torrens, 2010).
SABM are not confined to operate within a regularized data structure such as an indexed space partitioned grid, although they
could still be programed to do that. These two characteristics, freedom from central command and a good degree of
independence, combine to make a powerful modelling combination that has been successful in many domains to solve
complex problems. Generally, applications have been successful when spatial agents are designed to perform environmental
tasks such as map their surroundings or interrogate a complex space, monitor the state of things that may change over time or
simulate complex self-organizing systems such as anthills, bee's nests and traffic jams. For the purpose of this study, the
objective is to determine if agents can perform the initial three-dimensional graphical tasks that will be important for future
geological applications. The focus will be on visualizing and modelling local and regional anisotropy, and manipulation of
structural agents representing classic geology strike-dip and horizon-contact data.
**1.4 Role of Interpretation**
Earth Science in general, and geology in particular, is a domain characterized by the use of interpretation skills which are
fundamental to achieving successful practice. For problem representation, mapping applications and advancement of
knowledge in this field, experience and specific expertise is required to be able to solve complex spatial and temporal
relationships with limited observations (Brodaric, 2012; 2004). Knowledge of the processes that cumulatively produced the
resultant geometric forms, cross-cutting and overprinting relations and expectant natural patterns will drive an interpreter's
heuristic and narrow the solution space in which maps and cross-sections are developed. Ultimately for a reasonable three-
dimensional and four-dimensional model of the subsurface these interpretive skills are utilized to come up with a cohesive,
explanatory model that aims to reconcile and respect all the available data.
Spatial agents have the potential to support this interpretive role, provided some of their key characteristics can be leveraged
towards geological feature estimation and feature to feature relationship extension. This could be accomplished by more
efficient exploration of the model solution space through extension of horizon contacts, fault networks and fabrics.
**1.5  Demonstration Codes**
The properties and general behavior of spatial agents is demonstrated for the simplest of geological data, through several
agent demonstration programs. These codes and data can be freely downloaded (See
https://github.com/Loop3D/GeoSwarm.git or https://doi.org/10.5281/zenodo.4634021). The code implementation was done
in NetLogo 3D agent-based modelling software (Wilensky, 1999), taking inspiration from some earlier model examples such
as wave-3D (Wilensky, 1996) and flocking codes (Reynolds, 1987; Wilensky, 1998). The reader should download the
NetLogo 3D software and try some simple examples to gain a better appreciation of the agent environment (see Appendix A
for agent resources).  Each code example provided will have a NetLogo 3D implementation version that can run the code (see
Appendix B). Additional information to access the codes and a summary of the quaternion math specific for rotation and
interpolation of structural geology data used in this study is provided in Appendix C.
**2       Current Geological Surface Modelling**
Geological models are currently constructed through an iterative process of automated interpolation combined with
interpretation from data constraints (Caumon et al., 2009; Groshong, 2006).  Computer methods and workflows are applied to
data and output a collection of essential geological features, generally faults and horizons, which combine to form a
framework structural and stratigraphic model. When data is relatively abundant such as from three-dimensional seismic
surveys, common for hydrocarbon exploration and reservoir modelling, standardized methods do an excellent job at
representing sub-surface geological scenarios. However, when data become limited and geology more complex, precisely in
areas with high potential mineral, things can break down. In these circumstances existing implicit interpolation algorithms,
that are considered state-of-the-art for geology, may precisely fit the data but have much reduced global geologic accuracy.
See for example, figure 1 in which c) and d) are implicit geological  surface models developed respectively with
Gocad/SKUA (see https://www.pdgm.com/products/skua-gocad/) and SURFE radial basis function approaches (Hillier et al.;
2014). Note the missing representation of horizon C in the centre model c), and lack of through going spatial continuity of all
horizons in d). Both c) and d) would not be considered reasonable geological models by subject matter experts given the data.
Geological modelling is becoming a much more integrative, complex and computationally intensive undertaking (de
Kemp et al., 2017). There is a wealth of existing approaches for estimating geological surfaces with various data
types (geophysical, structural, stratigraphic) in a range of settings (Caumon et al., 2009). A common theme
emerging from the development of the arsenal of tools for this work, is that it is more and more difficult to come up
with a range of solutions that can both respect all the data inputs and the known complexity of features being
modelled (Jessell et al., 2014). In this under-determined problem domain, the move to leverage knowledge and data
to solve complex geology problems highlights the need to explore model spaces more efficiently for outcomes that
meet our minimum reasonableness criteria (Caumon et al., 2014, Jessell et al., 2014). Are agents a way to efficiently
tackle this problem, by providing a framework from which our existing tools can be embedded? This remains to be
seen, but at a minimum an exercise is needed to investigate if simple spatial agent operations can be used to model
structural geology data.
**2.1  Structural Agents**
This study focuses on the use of spatial agents for enhancing knowledge driven estimation, projections and
extension methods (Torrens, 2010; de kemp and Jessell, 2013) using sparse data, for regional geological domains.
Geoscience applications employing spatial agent-based modelling (SABM) have largely been focused on solving
time series problems, like land use change due to climate, urbanization and hazards (Torrens, 2010). Herein
however, the focus is on spatial variability, and distribution, rather than temporally changing environments.  The
major benefit of spatial agents is that they can be programed to act as a swarm. That is, they can act collectively,
having cohesion with their local neighbours, thus providing the spatial continuity required to construct continuous
features. The swarm may also be given shape-based rules, such as, keep members on a local plane or within a
specified degree of curvature. This is difficult to achieve with a global algorithm; inverting a matrix containing all
constraining data and properties. Spatial agents are potentially independent to explore a solution space that is not
constrained by regression minimizing criteria, which tend to make smooth solutions at the expense of realism.
Importantly, the cohesion of a swarm allows spatial agents to extend beyond the dense data regions, essentially
propagating features based on local rules, for example extending a surface along a fold plunge direction. Typically,
structural trends are manually traced in 2D, on maps and cross sections, with what are referred to as 'form lines' that
match the local planar fabric observations.  This can be done also in 3D, automatically (Hillier et al., 2013) but will
not provide feature continuity that the agents could provide. In the code examples, much use is made of what is
termed a 'structural agent'. These are agents that have spatial coordinate location properties for X, Y, Z but also
planar or linear geometric properties of strike, dip, trend, plunge and normal direction cosine components used to
designate a horizon top direction or a fold hinge line.  They may also have environmental information that tracks
local or regional eigen-fields. As noted earlier these types of agents may represent data, estimations or interrogators
that can transfer their properties as required. The structural agents enhance the interpretation process by densifying
the form lines and simulating more planar point features to highlight structural changes more clearly (Fig. 2).

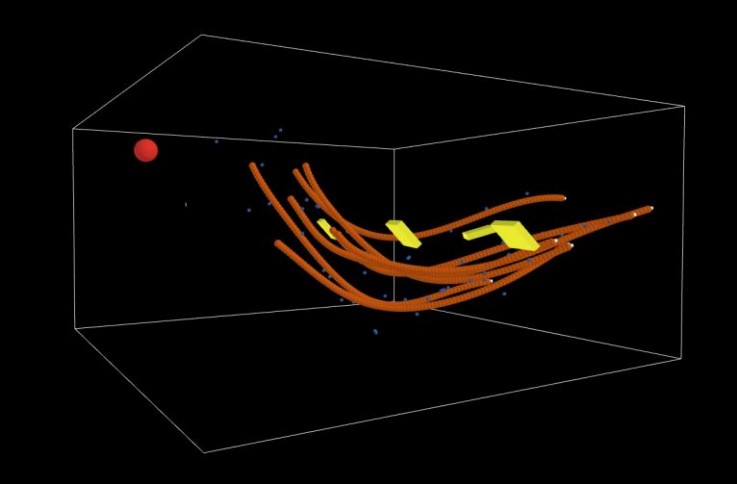

**Figure 2.** Structural form traces (orange point streams) estimated from dip data (yellow cuboids) using spatial agents. Red sphere
is an interrogator agent. Blue dots are simulated Bézier control points with added random noise. See Appendix A for details.
Spatial agents are also employed to better visualize and interpolate planar and linear structures, respecting the
polarity of the observations and resulting estimations (Fig. 3), essential for interpreting folded geology. Spatial agent
triangulated meshes are produced from point observations, that use proximity and topologic rules for accepting or
reject the meshing criteria to maintain local and overall continuity, meaning the surface has no holes or branches
(Fig. 4).  In this triangular meshing application, a random field of moving unconnected and randomly oriented
triangles is initialized. Each triangle is an agent set comprised of 3 node agents and 3D directed edges, as well as a
computed Barycentre with a unit normal vector property.  The closest triangle to the model centre will act as a seed
for the meshing and will sense its nearest neighbour triangle and connect to it, maintaining a consistent topology
with each triangle rotating into position, making a proper connection to an adjacent triangle. This proceeds until all
the triangles have been connected into a reasonable continuous surface patch, with no holes or large tears, and all
adjacent triangle normals pointing the same direction.  The action is very simple as shown in the pseudo code in
Appendix B.

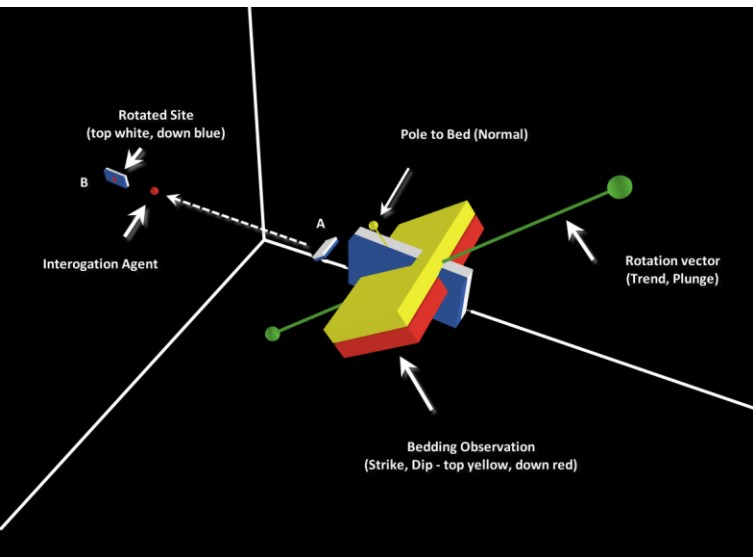

**Figure 3.**  Structural agent demonstrating a quaternion 90° clockwise rotation during linear estimation (SLERP) between two points. Starting
point A (local), with equivalent orientation to larger observation (yellow and red cuboid) and final rotated point B (distal). Rotation maintains
smooth topology for top direction. See Appendix B for details.

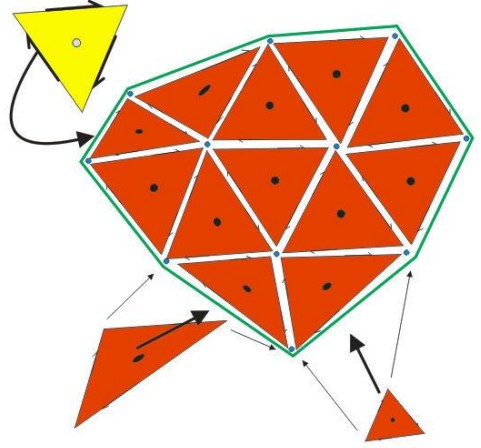

Inside Hull:

| Triangles | Mutual Edges | Shared Vertices |
|-----------|--------------|-----------------|
| 12        | 14           | 11              |

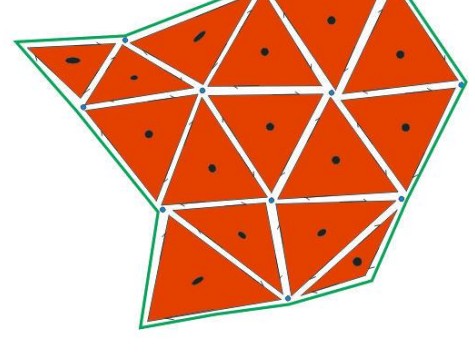

Inside Hull:

| Triangles | Mutual Edges | Shared Vertices |
|-----------|--------------|-----------------|
| 15        | 17           | 11              |

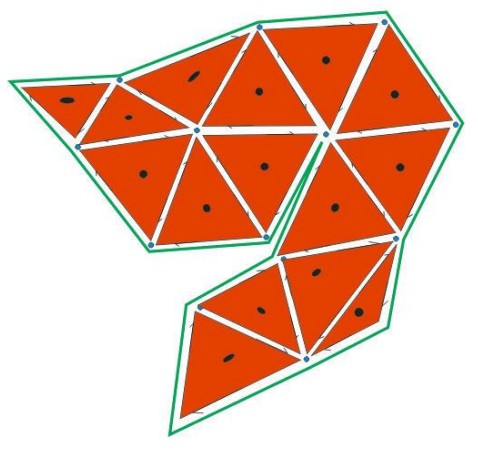

Inside Hull:

| Triangles | Mutual Edges | Shared Vertices |
|-----------|--------------|-----------------|
| 15        | 15           | 13              |

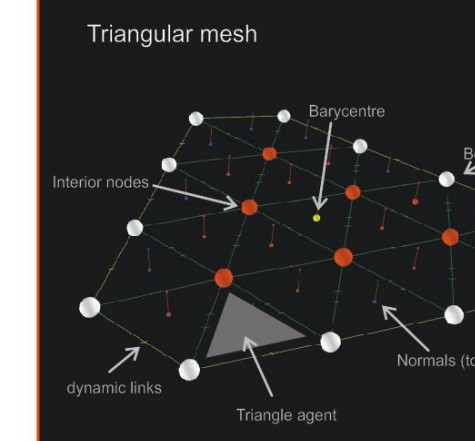

2      **Figure 4.** Spatial agent-based triangular meshing created from the Mesh program. See Appendix B for details.

**2.2     Agent  Communication**
There are a wide range of functions, behaviours and states that can be encoded into the agent set. These are collectively
driving what will be a successful application solution. Facilitating the efficient outcome of an agent model are agent
communications.  Inter-agent communication is handled through agent property updating (Fig. 5). Each agent is
responsible to know what is going on to the extent that it has been programed to, for example a proximity property may
be updated that indicates the nearest free agent neighbour, that is an agent not yet belonging to a swarm. Depending on
what behavior has been programed into the code, if an agent reaches a certain proximity threshold, an event might get
triggered such as to create an association link with that more proximal agent.

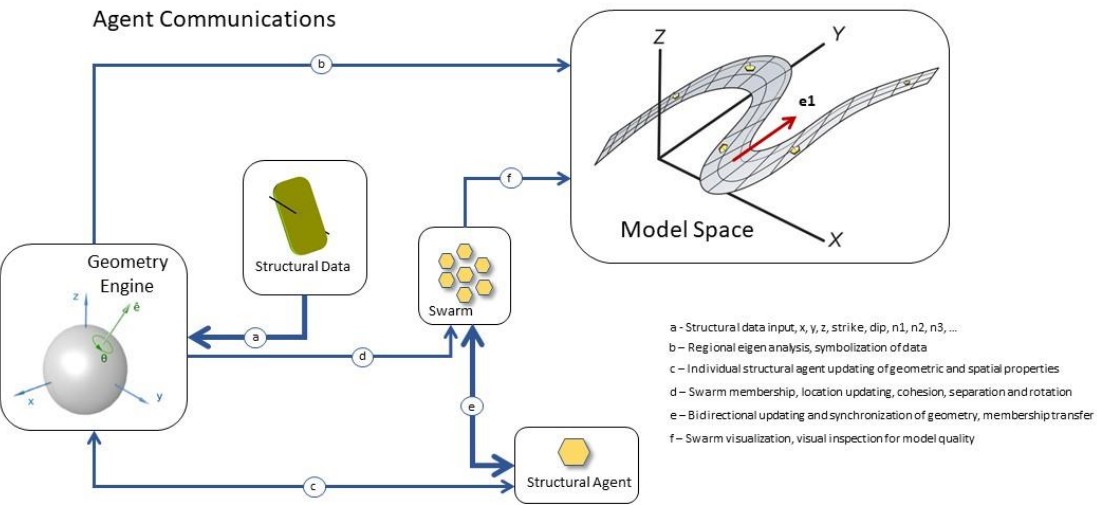

**Figure 5.**  General summary of structural agent communication using the example from *GeoSwarm*, for details see comments
and codes in the open-source programs listed in Appendix B. Thickness of blue lines indicates relative degree of inter-agent
communication. Geometry Engine composed of all agent functions for determining eigen directions, proximity, rotation, location
and spatial estimation. Grey fold surface in Model Space represents a possible fold realization that emerges from swarming
structural agents, given sparse input data (yellow markers). Red arrow indicates principal eigen vector direction, which is also
the fold hinge or regional plunge; this can be used as a rotation axis for structural agent geometry updating.
An agent can be made to act like an interrogator of space, whereby a continuous sampling may occur, in a given
direction rather than through a predefined set of indexed grid cells, such as in a convolution filter. Core to the behavior
of agents is the communication of derived weighting parameters for various properties, most importantly, for structural
orientation during interpolation. It is in this way that an agent can define a local neighbourhood as a local swarm, not
just by proximity, but also with geometric properties such as orientation. An agent might be very close to its neighbour
but may not be selected to be in the swarm because it is oriented at too high an angle thus promoting agents that are
near co-planar to be working together.  Agent interpolation is not actually replacing more classical schemes. SABM's
are more of a framework in which interpolation and other spatial operators can be called from as needed. Interpolation
schemes from simple to complex could be employed such as, nearest neighbour, inverse distance weighted (IDW) or
quaternion based spherical linear interpolation (SLERP) (De Paor, 1995; Shoemake, 1985; Hamilton, 1844).  Several
schemes could be employed depending on local or global data configurations, property conditions and knowledge
constraints. For the demo examples extensive use of SLERP methods ensure that rotations of geologic orientation data
are smooth and more realistic with respect to expected structural deformation processes.  In the presented examples,
there is yet no rheological controls, but these physical parameters could be programed into the agent rule set. Agents
can be programed to react to physical laws for example, the barycentre of a 3-tuplet mesh can be dynamically
recalculated when neighbour masses, other material and mechanical properties are changed. The location and states of
all agents are available and stored at the agent level, passed to a communications centre or just stored as a global
variable, if needed. Agent intercommunications is a significant topic of computational science research (Hall and
Virrantaus, 2016; Ménager, 2006), which may have implications for geological modelling, for example if moving into
the field of geological and geophysical integration and joint modelling, agents may have potential in optimization
strategies for inversion of complex geometries, multi-parameter scalar and vector fields (Jessell et al., 2010; Lindsay et
al., 2013). It is the way agents can communicate specific local to global information states, and adjust to the combined
data and knowledge constraints (Liscano et al., 2000; Friedrich et al., 1999; Gaspari, 1998), that may determine the
applicability of their use for geological and no doubt other applications as well. For a comprehensive summary of agent
and inter-agent communications and agent system controls see Heppenstall et al. (2012), for spatial agents with GIS see
Crooks and Heppenstall (2012), and for a practical introduction Wilensky and Rand (2015) (see also Appendix A).

### 2.3  Agent Behavior

Some interesting qualities of spatial agents:
**2.3.1**     Agents are able to efficiently interrogate irregular and complex model spaces. The model design can result
in a wide range of single realizations or solution suites. More traditional approaches are dependent on fixed regular
and partitioned structures using standard coordinate systems, with few geological properties.
**2.3.2**     Agents are suitable for modelling natural complex systems. Preserving contributions from multi-scalar and
deep multi-property data, such as fold shape parameters, or geophysical rock properties. Global interpolation
techniques such as implicit interpolation tend to generalize dense data clusters to a local mean and are optimized for
a scale specific purpose, often producing geologically meaningless results (Fig. 1). This could happen when
combining point geometry from structure, categorical geology, and continuous geophysics data. Essential details
such as fold topology and hinge regions can be ignored or conflict dramatically with geophysical gradients. Agents
may be able to more easily incorporate this kind of local information during estimation and feature propagation.
**2.3.3**     Agents can support the domain expert that requires more interpretive skills, with knowledge-based Rules,
Missions (Beliefs) and/or Behaviors during data interrogation. Agents could be used in mapping to visualize
complex relationships, such as within vector fields; for fabric intersections (bedding – cleavage relationships);
vergence relationships on fold trains; disharmonic folds and poly-deformed stratigraphy with early cryptic faulting.
Visualization of these relationships within the event history is critical to more accurate geological interpretation.
**2.3.4**     Agents complement rather than replace existing algorithms and approaches. For example, spatial estimation
can still be applied (Implicit, IDW- Inverse Distance Weighted, Kriging, DSI - Discrete Smooth Interpolation, SVM
Support Vector Machine, etc.) at variable scales as required. Thus, they potentially could provide a framework for
calling a variety of interpolators and constructors depending on data density, problem domain and feature
complexity.
**2.3.5**     Agent interaction and communication may produce group – swarm behavior. This emergence could
potentially express more complex features or trigger other spatial topological changes, such as new faults or
unconformities. Agents may also spawn, through their state condition, new geologic events altogether, for example
inserting a new deformation event when a metamorphic fabric is observed in a boulder of apparently undeformed
conglomerate, or when a high curvature region is detected by inserting a fold hinge or fault control point.
**2.3.6**    Agent-based approaches may benefit from denser and faster CPU/GPU architecture and parallelization
schemes. This could be the case, as the simple rules driving agent interaction and communication act more
independently, rather than having to invert large global matrices common in implicit approaches. This has yet to be
tested, since it is perhaps hard to partition on-going spawning processes from independent agents, but could result in
dramatic efficiency gains when combining multi-scalar properties from geophysics and geology within three-
dimensional structural fields (Burns, 1988; Hillier et al., 2013).
**3    Agents Examples**
To demonstrate the general principals of agent behavior for geologic surface development, a number of simple
applications were developed, using mostly synthetic data, and one re-scaled data set from an Archean greenstone
belt, Caopatina, Québec (de kemp, 2000), in a model space with (X,Y,Z) dimensions = (100,100,100) and model
centre at (X,Y,Z) = (0,0,0). The NetLogo codes presented are freely available for download (See
https://github.com/Loop3D/GeoSwarm.git or https://doi.org/10.5281/zenodo.4634021).
In the following example scenarios, spatial agents may represent control data, interrogators or estimated solutions.
They could also morph from one type to another. For example, a data agent could extend itself by expanding
incrementally along the dip plane directions into estimation points. They may have properties for tracking local
swarm or global states, continuously checking for proximity to neighbours, their status as interrogators or
observation sets and their geometric properties, such as strike, dip and polarity (top direction). Agents may have
pointers and links to specific topological neighbours as in the case of adjacent triangles but importantly there is no
ordered centralized control list, or matrix, which holds all the agents and their relationships. Each type of agent is
created and encoded with properties that may change, such as the local anisotropy derived from the eigenanalysis of
local supported data. The structural agents are spatial agents, represented herein as tablets or hexagonal glyphs and
rotate as quaternions (Fig. 6).

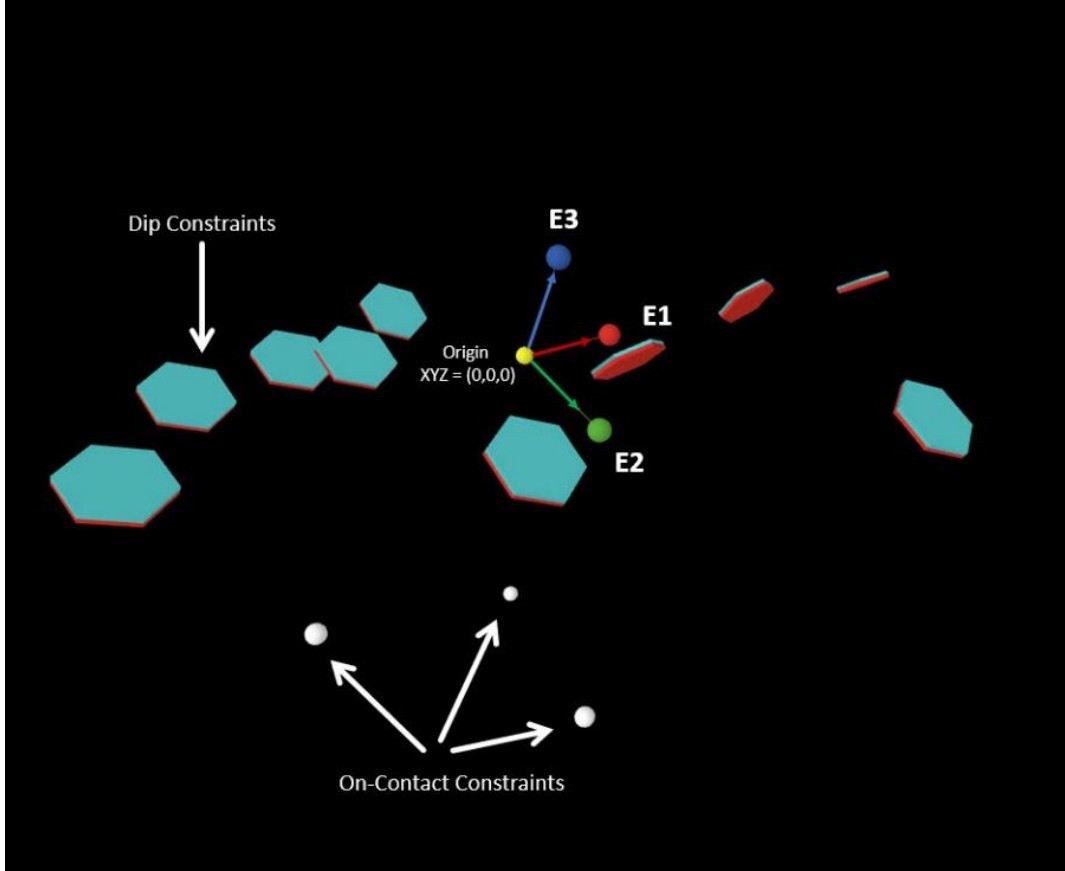

**Figure 6.** In-put Data constraints. On-contact (white spheres) and dip (blue=upright, red=down, thin hexagonal prisms)
representing simple three-dimensional geological data constraints. Arrows at origin indicate the calculated orthogonal unit
eigenvector directions for the structural data. Depending on the scenario the structural agents will do a SLERP interpolation (De
Paor, 1995; Shoemake, 1985; Hamilton, 1844) using a rotation vector from the major global eigenvector axis to simulate behavior
of bedding rotation due to near coaxial folding (Woodcock, 1977). For specific calculations used in each program see the code
comments or see Davis and Titus (2017) and the Appendix therein and Adamuszek et al. (2011), for a thorough review of structural
data computations. A summary of the quaternion rotation math is in Appendix C.

### 3.0.1 Scenarios

Each of the following programs runs inside NetLogo 3D, an agent simulation software which is freely available

from the Northwestern University NetLogo download site: http://ccl.northwestern.edu/NetLogo. The reader should

try the default parameters set when each program is called from NetLogo 3D and then adjust some of the simpler

parameters that control global orientation such as strike and dip. The descriptions below give the name of the

program, its intended behavior, and the main purpose of the demonstration code. Note that not all codes have been

thoroughly tested or gone through performance optimization. It is best to slowly increase the number of agent data
points for each scenario and experiment with the control parameters for best results.
*3.1     Trace*
Demonstrates the modelling of fabric observations (Fig. 2). The search agent (red sphere) travels through the model
space randomly until it senses a proximal dip observation. It will then adjust its trajectory towards a down dip vector
to this observation and spawn other simulated dip points that are nearest neighbour (NN) or inverse distance
weighted (IDW) interpolations from the data. A stream of points is recorded as the search agent moves through the
model space. This point stream will form De Casteljau – Bézier (Farin, 1997) curves that are either killed or
preserved based on simple user specified shape parameters, such as curve length. Other criteria have not yet been
implemented however this could be implemented, such as degree of curvature or mean direction angle from a
regional trend. Demonstrates streamline visualization using down dip trajectories. Similar to the three-dimensional
Structural Field Interpolation (SFI) from Hillier et al. (2013). The main distinction here is the sampling is random
with the potential for multiple search agents acting simultaneously.
*3.2     Poly*
Demonstrates simple polyhedral graphics control which is needed for vector-based boundary representations used in
many geological modelling environments. Construction agents can perform simple local tasks, such as making a
single polyhedron, but also regional tasks, by joining these up until stop-criteria are reached.  Modelling of simple
closed and connected polyhedra is achieved by joining simple triangles or large loops with many vertices. Each
closed polyhedron once formed will connect one link to its adjacent polyhedron, forming a simple object chain.
Modelling and visualization of the network are controlled by user-defined edge size, search radius, repulsion, and
tension of the edges.
*3.3     Rotate*
Demonstrates SLERP rotations, which would be required for estimation in complex geological domains, with
folding and sparse data representation (Fig. 3). It is also a testing environment for interpolating planar constraint
data with linear rotation axis. The main control dip agent is located at the origin in the centre of the model space and

a user defined target dip agent is set up. A linear quaternion rotation of the control dip is incrementally rotated along

a single or circular radial to the target dip. Users can rotate all dips continuously and dynamically. The agents are

always updating to the new target. Rotation axis is defined by the user which could be in all possible in-plane or out

of plane cross-dip orientations.   This is a required method for estimation of local and regional dips and structural

vector fields.

### *3.4     Mesh*

Demonstrates the development of topologic surfaces that, at a minimum, are defined by a triangulated mesh that has

direction and polarity sensitivity (Fig. 4), also to show that a mesh can be produced from agents without a grid;

without having to sample a scalar field value in a partitioned grid (i.e. with marching cube) and that meshes could be

grown locally, while conforming to constraint data. Each triangle has a normal that is maintained from the

barycentre of the triangle. Triangle vertices have a mass that can be changed by the user to influence the location of

the barycentre. A seed triangle senses the nearest neighbour triangle vertex and attracts it, back to itself. The

incoming triangle is rotated to be conformable to the evolving surface patch and connected, keeping the normal

pointing in the same way, thus maintaining simple surface topology. In this way distributed primitive shapes could

act as spatial data interrogators, before being transformed into mesh constructors. Simple topology metrics (edge:

vertices: triangles ratios) are reported and plotted on the GUI graph.  Once the mesh is complete, and if the on-

contact constraints are active, the mesh will migrate with its regional barycentre to the nearest on-surface control

point, and turn it blue from white, then go on to do the same for the next control point.  This functionality is a

precursor requirement for adaptive meshes, that could potentially be shaped by various spatial and property data,

data quality and data densities.  In this instance, a surface mesh is grown through use of simple geological rules. For

example, a surface can not intersect itself, and needs to be continuous with consistent surface polarity, and also to

avoid large tear faults. These surfaces may move toward on-contact data constraints to extend the local observations.

The ratios of triangles to shared edges and shared vertices can be used to check topology and used as a stopping-

criteria, to reward or penalize during the meshing process.

### 3.5.1 *Swarm Dips: Simple Plane*

This program demonstrates convergence of a non-meshed swarm toward a common plane. It is useful to
demonstrate proximity, vision distance effects, angle of sight and separation. Randomly initialized interrogation
agents, represented as smaller hexagons are dynamic, sensing agents and used to estimate or simulate, local
structural vector fields, herein referred to as Dip Sims. These Dip Sims slowly behave as a swarm, moving in the
plane specified by the controller, respecting vision-proximity and view-angle rules. When the separation and vision
distance are low, the sims will converge and produce red balls alerting the user that a proximity threshold has been
crossed. The red balls disappear once the sims move apart, and the inter-sim distance is greater than the specified
separation. This mode uses a single main dip controlling agent, represented by a large origin (0,0,0) centred, two-
sided (yellow up/green down) hexagon (see Fig. 7). The displayed data for on-contact and stationary dip data have
no influence. Only the main controller, large green-yellow hexagon symbol that is stationary at the model centre
with orientation (strike, dip, polarity) defined by the user, is influencing the swarm. The controlling parameters are
adjusted dynamically during the simulation run, initiated by pushing first the setup, and then the simulate buttons.
Dip Sims sense other Dip Sims within the vision distance and the view angle ($\phi$), they are kept from each other by a
user defined separation distance (yellow circle). The user changes the configuration during a simulation with sliders
on the NetLogo interface to control strike and dip properties of the Main Dip, which in turn controls the plane upon
which agents are moving on. The data in all the swarm examples are generated artificially by randomly positioned
sites on the plane of the main controller. The orientation of each dip data point is set by random rotation
perpendicular to the E1 (eigen) axis, to achieve a user specified variability (0 = no dip variance and 1 = maximum
dip variance). The idea is that each agent can see other agents within a locally controlled environment such as a
given vision distance and angle of sight, and these other agents start to coalesce forming a swarm, that could
potentially have some task to complete; extending a geologic feature of interest, extending a depositional horizon,
for example.

### 3.5.2 *Swarm Dips: Moving Plane with Dips*

Demonstrates smooth linear interpolation using SLERP (Spherical Linear Rotation Interpolation) with quaternions.
Parameterizes the rotation with linear segmentation of straight-line distance to controlling dip data. As the Dip Sims
come close to static dip data control-points they will adjust their local orientation to match the orientation properties
of the data, but do not move spatially towards these off-contact orientation observations (Fig. 7).

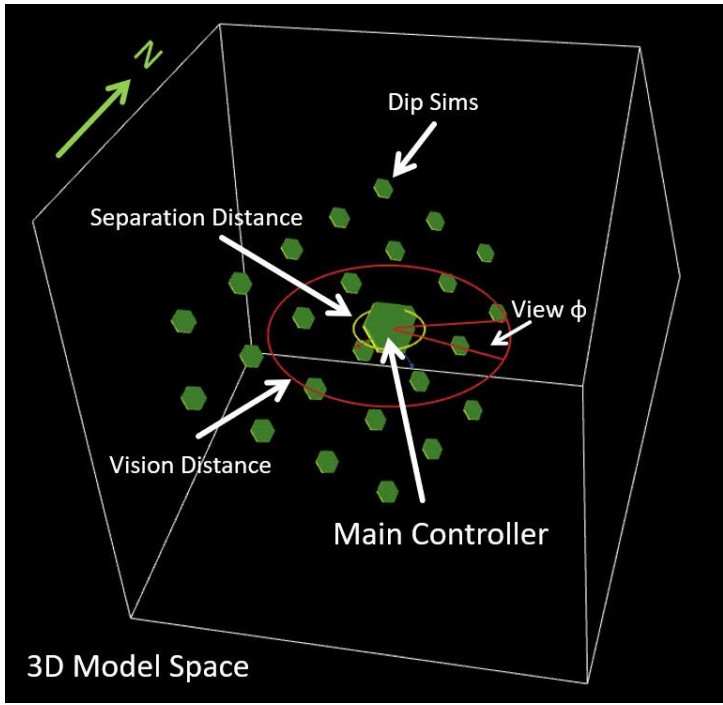

**Figure 7.** Components of the spatial agent-based model (SABM).
The influence of the orientation data on the estimation of orientation properties at the Dip Sim is weighted in an
inverse distance manner. There is no migration to on-contact data, only the off-contact dip data points have
influence. Outside the vision distance, the main regional controller determines the agent orientation.
*3.5.3    Swarm Dips: Migrate to On-contact Data*
Demonstrates that sims can sense and migrate to on-contact feature control points while detecting the structural
influence from adjacent data. Dip Sims move toward the nearest on-contact data point while rotating into parallelism
with the closest dip observation. At a given tolerance to the on-contact data points, the Dip Sim freezes in an
orientation that is close to the neighbourhood dip field. When all on-contact data points have a Dip Sim the rest of
the Dip Sims are behaving as a swarm; controlled by the Main controller and moving in the plane specified by the
controller and vision-proximity rules.
*3.6      GeoSwarm*
This example incorporates all of the above swarm methods using 4 separate structural observation files, or a random
set. The 4 test sets are taken from actual field data gathered from the Caopatina region, Québec, Canada, from
steeply dipping and folded series (Fig. 8) of turbiditic sediments from an Archean Greenstone Belt (de Kemp 2000).
Scaling settings can stretch the extents of the data for testing local versus regional influences on swarm cohesion.
Several distance sensitive parameters determine how agents are weighted for local surface cohesion versus data
migration. A file I/O interface for testing various data configurations representative of common but simple geologic
fold scenarios. It could be adapted for custom data configurations and will be used in the future for parameter
selection training and testing with a range of real data sets.

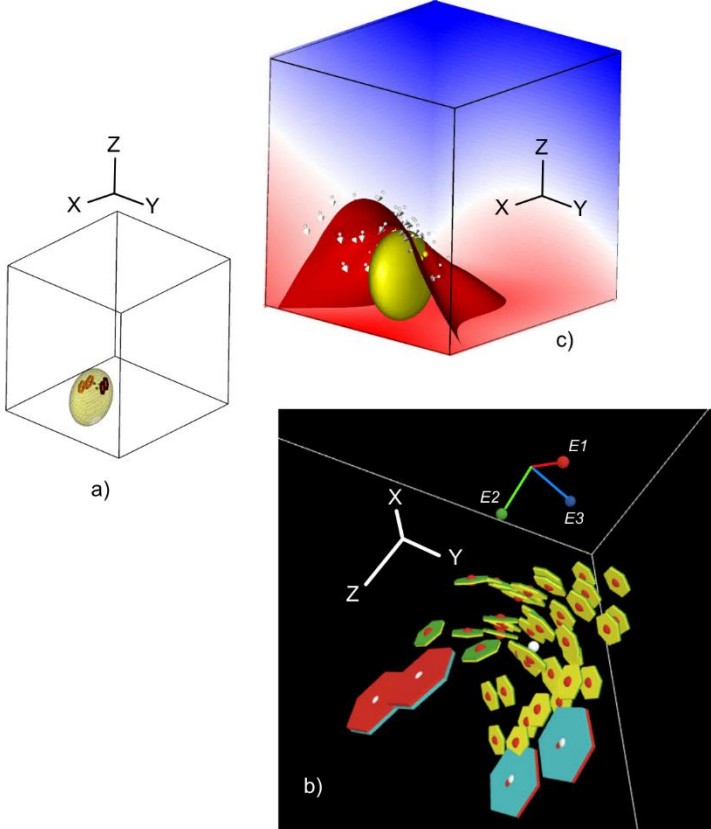

**Figure 8.**  Surface model (closed yellow ellipsoid) using implicit calculations with SURFE (Hillier et al., 2015) when using only
4 on-contact dip data points (a) and then using the GeoSwarm program to extend a fold plunge, with 50 off-contact spatial structural
agents depicted from the bottom, looking up in (b). Red surface in (c) is a more spatially continuous antiformal structure, when
using the structural agent approach than with implicit codes alone. Note eigen vector E1 (red stick-ball) is pointing down plunge
of the fold, the strongest continuity direction.
**4	Discussion and Conclusions**
This study focuses on the rudimentary requirements for geological modelling using spatial agents, primarily their ability to
interrogate, communicate and represent solutions to simple sparse geometric or structural constraint data configurations. No
doubt future research needs to go much further to see how to build full geological models, optimizing the arsenal of existing
geospatial tools within an agent framework.  Initial indications are promising for use of agents to develop meshing tools,
topologically sensitive surface construction of objects and for respecting simple geological data constraints such as on-
contact and dip observations.
The use of eigenvectors to summarize local anisotropic conditions derived from dip populations was helpful in supporting the
propagation of agents, weighting of the spatial continuity direction in a more intuitive manner for structural geological
interpretation, and selection of rotation axis for quaternion interpolations. These techniques, more commonly used in the
graphics industry, would be beneficial going forward in three-dimensional structural geological modelling in general and
potentially for more elaborate spatial agent approaches when solving for multi-property anisotropies such as occur in natural
geophysical and geological property distributions (De Paor, 1995). Sparse data configurations with more structural
variability, (see Fig. 8) when supported with an agent approach, will better reflect, and extend local structural anisotropy
when modelling using other methods such as with implicit estimators.
With the abundance of machine learning tools currently available it would be potentially useful to investigate how to
optimize structural agents for particular geological use cases, for example using self organizing maps and generalizations for
up-scaling structural data sets based on sampling from Kent distributions  (Carmichael and Aillères, 2016) for regional three
dimensional modelling or with application of graph neural networks for more complex geological modelling with sparse data
(Hillier et al., 2020) as well as other emerging deep learning approaches (Guo et al., 2021; Zhang et al., 2019).
Natural examples of agent behavior, such as swarm behavior, have emerged over millennia through the embedding of simple
rules into organisms that have evolved for optimization of their group survival. This paradigm, although perhaps not obvious
for geological applications, could take a similar path and could be an opportunity to leverage geological knowledge through
embedding of specific behaviours for given geological processes that are controlled through simple geological rule sets, for
example, by programing agents to maintain a range of thickness between stratigraphic layers as they are propagated

regionally.  Importantly, geological agents would need to operate in a geologically reasonable framework, respecting the local or regional geological topology network (Thiele et al., 2016). They would need to be able to create solutions from a suite of possible geological topologies with more complex feature sets, for example from combinations of geologic contacts and over printings, such as from horizons, faults, ore bodies, intrusions, alteration, and metamorphic fabric relations.

From this study it is clear that spatial agents can be used to develop simple meshed surfaces, fabric traces, visualize anisotropies and structurally sensitive swarm surfaces. Structural agent interrogators exploring a model space can update local or group behavior to conform to on-contact or within volume topological dip constraints.

Agent-based tools as applied to geological applications are yet in their infancy but can be used to interpolate or extrapolate from data to produce fabric trajectories, gradients, vector fields and continuous or discontinuous polyhedral meshed surfaces. The amplification of local anisotropies is particularly useful with sparse data and increased structural complexity scenarios. These characteristics can provide support for simulated input using existing methods for spatial estimation, such as implicit approaches.

Finally, more in-depth investigation into the use of and optimization of spatial agents needs to be undertaken to demonstrate the range of benefits for complex geological modelling in a variety of data configurations that could represent typical geological scenarios.

**Code and Data Availability**

These codes and data can be freely downloaded. (Please see: https://github.com/Loop3D/GeoSwarm.git or https://doi.org/10.5281/zenodo.4634021)

**Video Supplement**

The video files (mp4) related to this article are available online. (Please see https://github.com/Loop3D/GeoSwarm/tree/master/Docs or within https://doi.org/10.5281/zenodo.4634021).

**Author contributions**

EdK developed the GeoSwarm system, performed the literature review of spatial agents and wrote the paper.

**Competing interests**

The author declares that there is no conflict of interest.

**Special Issue Statement:** This contribution is part of the Loop stochastic geological modelling platform – development and applications, edited by Laurent Ailleres.

## 1 Acknowledgements

This research is part of and funded by the Canada 3D initiative at the Geological Survey of Canada.  Thanks to all the LOOP team (https://loop3d.github.io//  Australian Research Council: LP170100985) especially Mark Jessell and Laurent Ailleres for their patience and support of the project. Thanks to RING https://www.ring-team.org/ for academic support for use of Gocad/SKUA software and to Guillaume Caumon of who provided critical feedback for the research. Many thanks to Kevin Sprague who was inspired with the original notion of using agents for 3D geological modelling at the Geological Survey of Canada. Many thanks to Uri Wilensky, NetLogo team of developers and contributors who continue to enhance applications and extend the functionality of NetLogo with extensions to 3D, GIS, numerical functionality and other shared codes. Several examples such as wave and flocking codes have been the starting codes for this work presented here. Many thanks to Sarah D'Ettorre (2013) who initiated the first agent meshing codes now incorporated into Mesh.nlogo3d. Thanks to Doron Nussbaum, Carleton University, School of Computer Science who provided supervision of D'Ettorre while working on her MSc. Thanks to Mike Hiller and Boyan Brodaric (GSC), for valuable feedback. Early consultation on AI methods from Éric Beaudry, (Université du Québec a Montréal), Khalid Djado and Mathieu Bouyrie (Kinna Technologies) is appreciated. Thanks to an anonymous reviewer and Guillaume Duclaux for comments and suggestions, which greatly improved the manuscript. NRCan Contribution Number xxx.

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

**Appendix A:     Agent resources**
An excellent starting point to become familiar with agent-based applications and approaches is Paul Torrens' web site at
http://geosimulation.org/ from the Computer Science and Engineering, Tandon School and Center for Urban Science and
Progress, at New York University.
The agent-based codes used in this paper are written in Net Logo-3D, a spatial agent-based modelling language and
development environment that is supported from the Center for Connected Learning and Computer-based modelling in
Evanston, Illinois, USA. The NetLogo project is affiliated with Northwestern University Centre on Complex Systems
(NICO) https://www.nico.northwestern.edu/ . To download and run the NetLogo codes, for tutorials and documentation on
the NetLogo language see http://ccl.northwestern.edu/NetLogo. The code must be minimally compatible with the NetLogo
3D  version as listed in the programs below. Current and early 3D versions of the program are all available on the main
NetLogo homepage.
Codes presented in this paper are freely downloadable from the Git Hub Open Source web site at
https://github.com/Loop3D/GeoSwarm.git (https://doi.org/10.5281/zenodo.4634021) with accompanying power point, pdf
and animations presented at the annual meeting of the International Association of Mathematical Geoscientists at Penn State
University, USA, August  2019.

1    **Appendix B:**          **List of NetLogo 3D Programs**

2    *Program Name*          *Version*          *Purpose*

| Program Name | Version | Purpose |
|---|---|---|
| 3 | Trace.nlogo3d | 6.0.4 | Propagation and interpolation (NN and IDW) |
| 4 | Poly.nlogo3d | 6.0.4 | Closed and connected polyhedral growth |
| 5 | Mesh.nlogo3d | 6.0.4 | Simple surface meshing by triangulation growth |
| 6 | Rotate.nlogo3d | 6.0.4 | Dips with polarity rotation (SLERP - eigenvectors) |
| 7 | Swarm_Dips.nlogo3d | 6.0.4 | Structural dip cohesion mimicking deformed surfaces |
| 8 | GeoSwarm.nlogo3d | 6.0.4 | Simple geometry solving from steep fold limb pairs |
| 9 | Wave.nlogo3d | 6.0.4 | Simple non-meshed elastic surface motion |

11    *Shape Libraries:*

| | Library | Version | Purpose |
|---|---|---|---|
| 12 | 3d_HexShape.txt | > 5.3 | Required to generate hexagon dip glyphs with polarity |
| 13 | **3**d_Shape.txt | 4.1,5.1,6.0.4 | Required to generate tabular dip glyphs with polarity |

**Example Pseudo Code:**

**Mesh.nlogo3d**

```
Start
Create Nodes agent set
Create Triangles agent set with random directed normals
Define a seed Triangle
Do while [ mesh growing ] [
if [nearest neighbour to seed Triangle exists] [
connect an edge of the seed Triangle to its nearest neighbour's edge
repeat along the seed until all its edges are fused
repeat along the outer edge of the mesh
]
if [all Triangles meshed] [
quality check the mesh
if [mesh is not reasonable] [
set mesh to growing
disconnect the mesh by killing shared edges
Scatter all Triangles
Re-define the seed Triangle
]
Else
```

1                                  [set mesh to not growing]

2                               ]

3                       ]

4          End

Once all the meshing is complete, there is a quality control check to determine if the result is a 'reasonable' surface.
This could be a simple rule that looks for holes, and surfaces with low connectivity, for example by calculating a
low node count to edge count ratio; with 1 = no triangles connected, ~ .72 = single node connected chain, ~ .62 =
single edge connected chain, ~ .58 = hexagonal mesh).
**Appendix C:    Quaternion Calculations**
Quaternion codes are used in Dip_Swarm and Rotate programs and implemented in NetLogo within the **Spin()**
procedure.  Used for smooth rotation along specified axis such as an eigenvector of a structural observation set and
for inverse distance weighted (IDW) and Spherical Linear Interpolation (SLERP). For details see De Paor (1995),
Shoemake (1985), Hamilton (1853).
**C.1**      Provide a normalized unit vector to the Spin procedure from common structural observation data
Convert strike and dip (RHR) to a Unit Normal vector. Input is in degrees. Normal is perpendicular to plane
*strike* = a scalar angle of in degrees azimuth in the horizontal plane measured clockwise from north (0º) representing
the angle between a topographic surface trace of a geological feature, such as a horizon intersecting with
topography, and the north direction. Strike in this study uses the Right Hand Rule (RHR) which is a common
structural geological measuring standard for planar field observation data. It assumes that the strike direction vector
is pointing such that the geological surface dips to the right of the observer as they face the strike direction.

13            ( Note east = 90º, south = 180 º, west = 270 º)

*strike* ∈ {0,360}
*dip* = a scalar angle in degrees indicating maximum slope from the horizontal taken in the direction of the dipping
surface. The dip direction is always 90º to the strike direction. The dip angle (dip) is the maximum vertical angle
from the horizontal to the geological surface.
*dip* ∈ {0,90}
*polarity* ∈ {−1,0,1}
polarity = a signed unit integer indicating if a geological surface is upside down, that is overturned with respect to its
original depositional configuration. -1 = overturned, 0 = unknown, 1 = upright. This value is used to give topological
information in modelling.
strdip2norm ( *strike, dip, polarity* )
Returns a 3 element unit normal vector.
Calculate down dip vector

$ddx = \cos(-1 * strike) * \cos(-1 * dip)$

$ddy = \sin(-1 * strike) * \cos(-1 * dip)$

$ddz = \sin(-1 * dip)$

Calculate the strike vector

$sx = -1 * ddy$

$sy = ddx$

$sz = 0$  (note the strike vector is always in the horizontal plane)

Cross down dip vector with strike vector ($V_{dd}$ X $V_s$ to get the normal ($N$) or pole to bedding.

$NNx = (ddy * sz) - (ddz * sy)$

$NNy = (ddz * sx) - (ddx * sz)$

$NNz = (ddx * sy) - (ddy * sx)$

Normalize the normal for unit length *L*.
$L = \sqrt{NNx^2 + NNy^2 + NNz^2}$
Adjust for polarity
$Nx$ = (polarity * $NNx$) / *L*
$Ny$ = (polarity * $NNy$) / *L*
$Nz$ = (polarity * $NNz$) / *L*
Convert a Trend and Plunge to a normalized unit Vector. A common fabric element for various linear structural
features such as fold hinge lines joining maximum curvatures along the plunge of a fold, or stretching features
located along E3. Used to get a vector from an agent heading and pitch state.
TrendPlunge2Vec (*trend, plunge*)
Returns a 3 element unit normal vector.
$VVx$ = sin ( trend ) *  cos (plunge )
$VVy$ = cos ( trend ) * cos ( plunge )
$VVz$ = sin ( plunge )
Unit Normalize
$M = \sqrt{VVx^2 + VVy^2 + VVz^2}$
$Vx$ = $VVx$ / *M*
$Vy$ = $VVy$ / *M*
$Vz$ = $VVz$ / *M*
**C.2**     Input the rotation increments (A) the rotation vector (**Q**) and the normal of the structural observation (**P**)
into the Spin procedure to rotate the structural elements with quaternion calculations.
**Spin** *( A V P )*
*A* = spherical angle of rotation in degrees (not Euler angles) $A \in \{-\infty, \infty\}$
*V* = Unit vector 3D axis of rotation (Vx, Vy, Vz,). Can be any of the eigenvectors, a down dip vector, strike vector
etc.
*P* = Normal unit vector ($n_x$, $n_y$, $n_z$)    (such as  Poles to beds, a fold hinge etc. )
Returns *S* a matrix with full orientation description including the normal to bedding or new rotated linear element,
the *strike* and *dip* components, *overturned* (polarity) and 4 quaternion elements (*qw, qx, qy, qz*).
Transform from single vector to quaternion with rotation *A* about an axis **Q**
This procedure can be used to convert normal to strike and dip RHR by input *A* = 0 rotation and *V* = *P* just cast the *P*
as a single matrix from the normal
Returns RHR_Orientation array using Right Hand Rule planar orientation for STRIKE, DIP, N1, N2 ,N3,
OVERTURNED
**Q** = (*s, V*)  scalar , vector
$qx = (\sin(A/2) * Vx$
$qy = (\sin(A/2) * Vy$
$qz = (\sin(A/2) * Vz$
$qw = (\cos(A/2))$
**Q** = (*qw, qx, qy, qz*)
**C.3**  Create the Rotation Matrix
Use quaternion identities to derive the rotation matrix
$q2w = 1 - qx^2 - qy^2 - qz^2$
$q2x = qx^2$
$q2y = qy^2$
$q2z = qz^2$
Compose *R* the rotation matrix
$R = \begin{matrix} q2w+q2x-q2y-q2z & 2qxqy-2qzqw & 2qzqx+2qyqw \\ 2qxqy+2qzqw & q2wq2x+q2y-q2z & 2qyqz-2qwqx \\ 2qzqx-2qyqw & 2qyqz+2qwqx & q2wq2x-q2y+q2z \end{matrix}$
$R = \begin{matrix} R_{xx} & R_{xy} & R_{xz} \\ R_{yx} & R_{yy} & R_{yz} \\ R_{zx} & R_{zy} & R_{zz} \end{matrix}$
**C.4**      Matrix multiply the Rotation matrix with the input observation normal P
*S* = *P X R*
**C.5**      Interpolate, by calling the spherical linear interpolator (SLERP) for any interpolation on parameter *t*, a
normalized distance between data and the spatial starting point of an agent (*A*) as it is rotated towards the structural
constraint (*B*). Details of SLERP can be located in De Paor (1995). Note with the inverse distance weighted (IDW)
form of SLERP a set of structures can all influence the agent depending on the agent's ability to sense the data, for
example the structural search agent needs to be within the vision distance.
For the IDW – SLERP calculate the data weights based on inverse distance, adjust exponent *p* if needed for stronger
local influence,
$$W_i = \frac{1}{D_i{}^p \sum_{j=1}^{n}(\frac{1}{D_j^p})}$$
Calculate *G* the estimated orientation at *x* by adjusting the contributing quaternion components of the data with the
distance weights,
$$G(x) = \sum_{i=1}^{n}(W_i * Q_i)$$
To use the simpler linear form with A and B orientations,
$$SLERP(x) = (1-t)Q_A + tQ_B \quad (t = |\,dist\,|)$$

