# Peer review of "Spatial Agents for Geological Surface Modelling"

_Geoscientific Model Development, 2021_

## Author Response (AR1)

**1 Authors Response to Refereed Comments for gmd-2021-66**

Dear GMD Referees,

I can not thank you enough for taking the time to do a thorough review of my paper 'Spatial Agents for Geological Surface Modelling'. Especially, since there are, as noted in your review intro's, aspects that you may not have been familiar with either from the agent side, or from the structural geology perspective. To be sure, the programing that was done in Netlogo is also not familiar to many researchers. It is an interesting proto-typing environment that is easy to learn and accessible to the public, so perhaps that may help in reaching the masses. Your comments are both appreciated, and I hope all dealt with, as outlined below, with an aim to increasing readability and clarity for a wider audience. I tried to do my best to address your concerns through re-organizing, adding definitions of terms, making key points early and adding some relevant references should someone pick up this research in the future. I also added a new figure (Fig. 5) summarizing agent communications that shows, in a basic way, how the main components of the system work.

- 12 I have not seen any other comments or concerns come in through the GMD discussion site, but perhaps there may be a need 13 to address things after reading this version and posting it on the GMD site.
- 14 Regards, trick. Idep

- 16 Eric A. de Kemp
- 17
- 18 Ottawa, Canada
- 19
- 20 Note Author Responses in (Arial Font) Dark Red (with Review Simple Markup)
- 21 Page and Line numbers from new document Spatial\_Agents\_GMD\_r5.pdf (Markup on).
- 22

**23 Comment on gmd-2021-66**

- 24 Anonymous Referee #1
- 25 Referee comment on "Spatial Agents for Geological Surface Modelling" by Eric A. de Kemp,
- 26 Geosci. Model Dev. Discuss., https://doi.org/10.5194/gmd-2021-66-RC1, 2021
- 27 General comments:
- 28 I'll start with a caveat: in retrospect, I am probably not be an ideal reviewer for this
- 29 paper, as I have no experience in the application of solid modeling techniques to infer and
- 30 visualize subsurface geological structures. However, a silver lining maybe is that I can
- provide a general geoscience perspective on this paper.
- This is probably a good thing as, I believe, it forced me to make things more understandable for the nonspecialist.
- 35
- 36 (One terminology note: the phrase 'geological modelling' is used here to mean creating
- 37 digital representations of 3D sub-surface geological structures, but the same term also refers to the use of numerical models to study the dynamics of geological and geophysical
systems. These are very different things, so a definition early in the paper would be
helpful.)

I have included a line in the abstract "...approaches to creating 3D geological models involves development of
surface components that represent spatial geological features, horizons, faults and folds, and then assembling
them into a framework model as context for down-stream property modelling applications (geophysical inversions,
thermo-mechanical simulations, fracture density models etc.)." to clarify this modelling definition.

11 In addition ....

P.2, L.24 ... Introduction "Herein we focus on the starting framework model, the stratigraphic and structural
 surface model that provides the initial context for these more down-stream property embedded modelling efforts."

This paper makes two valuable contributions. First, it evaluates the use of agent-based
modeling techniques, which are widely used to study system dynamics in fields like
ecology and sociology, for inferring and graphically representing 3D geological structures
from sparse data.

- 19
- 20 That is a good characterization of what I was trying to do.21
- Second, it presents a new open-source software package to carry outthis kind of modeling, using the NetLogo package.
- 24
- Yes, for sure and hopefully initiate future research with other agent environments.
- 27 To the best of my (admittedly limited)
- knowledge, both of these represent novel, interesting, and valuable contributions. And
  clearly a tremendous amount of effort has gone into developing the ideas and the
  accompanying software implementation.
- 32 Excellent, thank you for that!
- 33

- 34 For these reasons, I feel that the manuscript is
- appropriate for Geoscientific Model Development, and should find an audience among
   geologists who are interested in inferring subsurface geological structures from limited
- 37 observations.
- 38
- 39 Great!
- 40
- 41 One of the challenges with this manuscript is that it assumes a lot of background
- knowledge on the part of the reader, which risks limiting its impact. One recommendationtherefore is to do some fairly thorough editing to make it more widely accessible.
- 44
- I fully agree. Without pointing out all changes, the mark-up document in word is maybe the best
  way to go to check these changes. The Spatial\_Agents\_GMD\_r5.pdf content is also loaded, below, to
  see the extensive changes that have been made.
- 49 One thing that I believe would help is some re-ordering of the information presented,
- 50 especially in the introduction. Such re-ordering could make this manuscript accessible to a
- 51 wider audience, and therefore draw more attention to this important work. Here's one
- 52 potential re-organization of the introductory material:

The intro (P.2,L.15 to P.9,L5) has been expanded, with more sections and completely re-organized with lots of re-writing to make things clearer. I spell out more clearly what the problem is and the challenges we face in modelling, and importantly why the agents may be helpful in dealing with this. An early reference to Fig. 1 should help as it visually states the problem with standard models looking like nothing seen in geology.

"The *major challenge* that this paper is trying to address is the breakdown in achieving geologically realistic model results 10 from sparse data in more complicated geological scenarios when using the existing methods and algorithms. This is no doubt 11 a problem in other modelling domains as well, but is acute in geological applications, where access to data in the subsurface 12 is often extremely expensive, terrain access prohibitive, or the depth of investigation too extreme for direct sampling and 13 must rely on coarser geophysical methods that often do not adequately image the features being modelled. This paper 14 explorers the use of extension, propagation and cohesion methods, which can be considered part of 'swarm' technology, 15 using spatial agents in an attempt to deal with this challenge.

Geological modelling covers a wide range of applications and domains from thermo-mechanical modelling (Cloetingh et al., 2013) to basin analysis (Barrett et al., 2018), mineral potential estimation (Skirrow et al., 2019) in 3D (Hu et al., 2020;
Sprague et al., 2006) and even 4D applications (Parquer et al., 2020; White, 2013). *Herein we focus on the starting framework model*, the stratigraphic and structural surface model that provides the initial context for these more down-stream property embedded modelling efforts. "

a - Explain briefly what is meant by geological modelling in this paper, and why it matters
(a few sentences or a paragraph, with general references for those unfamiliar with the
field/area).

See above comments.

b - Explain the major challenges that this paper and the techniques and software it
describes are meant to address (e.g., sparse information about the subsurface,
uncertainty in 3D location and/or properties, long computation times using standard
algorithms, or whatever)

again, see above comments.

c - Explain why current methods are limited (i.e., why do we need a new and different
approach?), and thus why it's worth trying an alternative approach.

In P.3, L.7, I add an explanation as to why we need a new approach and agents may be theanswer...

This is dealt with also in a new titled section 1.1 Agent Challenge and later in (P.5, L.8) 1.4 Role of *Interpretation*, (P.8 L.10) which is really what we do now to deal with sparse data; we use our
knowledge either instead of algorithms, or to supplement them.

"Existing methods applied to the combined sparse data and complex geology scenario, will tend to produce holes, gaps and feature drop-outs, away from control data, as well as arbitrary horizon thickness changes that combine to give a geologically unreasonable bubble gum look to these models (Fig. 1). Current methods in sparse data configurations tend to bias for these unrealistic geometries using radial based kernel functions, optimized for local smoothness in order to achieve
a mathematical solution (Hillier et al., 2021; Hillier et al., 2014). This often comes at the price of geological realism (Hillier et al.)

a mathematical solution (miller et al., 2021; miller et al., 2014). This often comes at the price of geological fealism (Hiller et al., 2021; MacCormack and Eyles; 2012). Is it possible that, with a new approach, geological features could be more

- 50 realistically modelled by using spatial agents to 'fill-the-gaps' in the process?"
- 51 52

d - Give a quick background summary (2-3 sentences) on agent-based modeling that
gives readers a basic sense of what it is, e.g., that it's a technique used in simulation
modeling of complex systems in which individual entities such as animals or households
interact with an environment and with one another.

This has been dealt with in Intro Section 1.2 (P.6, L.3...) 7

"In general, an agent-based system is used to see the effects of autonomous individuals, groups or objects on the overall
system when solutions are onerous and/or computationally expensive. A global algorithm involving a single large multi10 parameter matrix inversion may take many days to compute with a single outcome, but an agent-based model may be able to
produce several outcomes in minutes or hours (Siegfried, 2014). Agent-based models have their roots in the development of
cellular automata and complexity theory, which has been able to model complex natural and artificial systems with simple
neighbourhood algorithms (Cervelle and Formenti, 2009; Wolfram, 1994; Von Neumann, 1966)...."

15

e - wrap up the intro with a quick summary of what this paper does (maybe this is a good place to say you're doing this in the context of Loop 3D, and explain briefly what that is).
This way, by the end of your Intro section, readers will know what problem you are trying to solve, why it matters, why it is a problem, and what (basically) your proposed approach is. You have all of the pieces already, but they are currently presented in an order that risks leaving readers drowning in a sea of jargon before they have a chance to get to the cool new ideas and techniques.

Agreed, to much jargon. All the pieces have been re-ordered. It flows a bit better, in my opinion.
The end of the intro (P.3, L.7-24) is much improved and clearer for the general modeller.

The general Intro is followed by sections 1.1 to 1.5. More details and explanations were added in
these sections. I added two references from Brodaric on interpretation that will give some
background on geologic mapping practice. 3D modelling is just an extension of this 2D practice
that requires mental extension and propagation functions from experience of natural forms. We
want to capitalize on some of these functions with agents. (Cohesion, extension etc.)

- 3233 (P. 5, L.1) Section 1.1 Agent Challenge
- 34 (P. 6, L.3) Section 1.2 Agent Applications
- 35 (P.7, L.13) Section 1.3 Agent Characteristics
- 36 (P.8, L.10) Section 1.4 Role of Interpretation
- 37 (P.8, L.22) Section 1.5 Demonstration Codes38

These could all be wrapped into a section 2 and the rest of the sections incremented, but I felt each
of these sections was just an intro into the topics that do get more elaborated later in the paper.
This could be changed, perhaps it would flow better?

A related challenge is that the manuscript does not really articulate (at least not until deep
into the details) what is wrong with current modeling algorithms and why an agent-based
approach might offer a better alternative. A couple of sentences articulating this
somewhere in the introduction would help motivate the rest of the paper.

- 47
- 48 This is now up front in the intro ...
- 49 (P. 5, L.1) Section 1.1 Agent Challenge
- 50

One general grammatical note: the manuscript contains quite a few incomplete sentences, which need to be fixed before final publication. I have flagged some of these below with

'inc'.

Thanks for taking the time to look at these. These were all fixed. I do have some kind of mental
block with starting sentences with a conjunction; such as, for example ...

Specific comments (by page and line numbers):

1, 13-15 this opening statement of the abstract is a nice example of the challenge readers 8 face with this piece (but there is a potential fix). For a reader to care about the topic in this sentence, they have to know what is meant by 'spatial and property interrogation 9 functions' and 'estimations and construction operations'. It's only at the end of the 10 11 sentence that we get to the heart of the matter: 3D geological surfaces. You have an opportunity to invert this in a way that could make it more accessible. Start off with a 12 statement about how important it is build 3D-rendered models of geologic structures, and 13 how the sparsity of data makes this difficult. Then state that agent-based modeling 14 15 provides a potential solution, and that the contribution of this paper is to test it out. By laying out the ideas in this sequence, right away you have presented an interesting 16 research challenge, and followed up with a solution. Now the reader has a reason to care 17 18 about technical things like property interrogation functions.

Absolutely. I was too close to the trees on this. The abstract is re-written with a new paragraph
added, to focus on the problem that the Geological surveys face, in-fact my own work practice
faces this every day and a major reason I started going down this road. The abstract is still under
words.

P.1, L.13 "Increased availability and use of 3D rendered geological models has provided society with predictive
 capabilities, supporting natural resource assessments, hazards awareness and infrastructure development. The Geological
 Survey of Canada, along with other such institutions, have been trying to standardize and operationalize this modelling
 practice."

1, 18-19 geologic modelling has multiple meanings - please define what you mean by it inthis context.

See comments above. Dealt with.

2, 3-22 As noted above, the introduction is hard to follow. It could benefit from a clearer
articulation of the nature of the goal and the need for solutions. In particular, it sounds
like the core problem is one of limited data, so why does the answer lie in software
architecture? How do those two things relate?

Again, see comments above on the restructuring and enhancement to the intro with a new section 1.4 Role of Interpretation. The limited data and software architecture relationship is dealt with 41 42 (P.8,L.19) by showing the link between knowledge driven map interpretation (drawing lines on 2D surfaces with some trend geometry data points) in 2D, and ability of agents to links features 43 sampled with sparse data. I do not make an elaborate final application that does this, but I think 44 45 there is enough there to demonstrate that with all the pieces, in this agent framework, it could be 46 done. Down the road, I think especially the cohesion and eventually rheological (stiffness; response 47 to stress; fold characteristics etc.) properties of agents could help here.

"Spatial agents have the potential to support this interpretive role, provided some of their key characteristics can be leveraged towards geological feature estimation and feature to feature relationship extension. This could be accomplished by more efficient exploration of the model solution space through extension of horizon contacts, fault networks and fabrics."

2 2, 23 - 3, 1: I have some familiarity with agent-based modelling but I worry that readers 3 who have never heard of it would struggle to get the basic idea from this somewhat abstract description. Maybe it would help to start off not by identifying different categories 4 of ABMs, but instead to convey the gist of the technique first and what fields it is used in 5 (which you address later in the MS; e.g., to simulate individual entities in computer 6 7 games, animals in ecological simulations, households or sectors in economic simulations, 8 etc.) I do like the carpenter example (examples are always helpful) but in that particular example it's not clear why 'single agent' applies (is the carpenter one agent, and the 9 house another? is there only one carpenter?) 10

Yes, agreed. I move the general agent introduction and explanation earlier in the paper. 13

3, 8 Surely ABMs can themselves be computationally expensive. It is not clear why they
would be expected to reduce computation time relative to whatever the alternatives might
be (presumably some kind of 'global' algorithm?).

I include a reference from Siegfried (2014) to support this, see his text intro on 1.1 Motivation
where he argues that for "Solving complex systems ... characterized by non-linear aggregate behavior (i.e. The
aggregate behavior of the individual components is not derivable from the summation of the activities of individual
components)" and he goes on to say that complex natural systems are essentially requiring a simulation approach
to model in a meaningful way. I think the structural agent framework is just that kind of solution.

I agree, for me it is more of an intuitive notion and yes, I am leaping a bit. I have seen firsthand what matrix 25 solvers (large multi-parameter inversions) for geoscience data (geophysical, structural inversion) can provide. It is 26 far from satisfactory, and is like trying to tie thousands of elastics with various strengths (weights) inside a multi-27 sided container with movable walls. You may eventually (after a very long time) get the container to be stable and 28 stand on its own, but it will look nothing like it was supposed to represent. With agents there is no balancing act, 29 to invert a large mathematically conditioned matrix, that may or may not be solvable. Instead, we allow the system 30 to do simpler calculations, with simple rules at the local level. Perhaps a solution, or similar but many solutions, 31 emerge depending on how well we have designed the system. It does seem intuitively more natural and more 32 efficient to do this. This has yet to be rigorously tested, which I have only scratched the surface of in this study. 33 That is why I wanted it to be a more concept paper.

3, 15-18 This is a useful statement of the paper's objective. It sets up readers to expect to
learn next what are these graphical tasks, and why agent-based techniques makes them
easier or more efficient or more effective, etc.

Good.

3, 20 - 4, 2 This is a nice overview of ABM applications. Consider moving it before the
more abstract discussion of agent properties. It gives readers a sense for how and why
the technique is used.

Yes, the section was moved to an earlier part of the paper.

4, 3-15 It is not totally clear what you mean by anisotropy and gradient-type information.
Consider leaving that bit for later, when the application examples will make it easier to
understand in context. More generally, there is a lot of jargon in this paragraph: to make
sense of it, a reader has to know what you mean by multi-scalar environment, model
element interactions, multi-source physical dependencies, non-centralized control
structures, global partitioned data structures, etc. Consider either adding text to define
these various terms, or deleting them, or replacing them with more accessible descriptions. That said, the examples are great; in general, more examples and fewer 2 jargon terms would help. 3 Definitions for Anisotropy, gradient-type and jargon clarified. 4 5 See P.5, L.20 "better model the local structural trends or anisotropy, and extend features such as regional fold 6 plunges." 7 8 Intro rewording and added examples to help clarify terms along with multi-scalar use is brought 9 out in the example in the intro of molecular to galactic modelling. 10 (see P.6,L.20 to P.7,L1) 11 12 "These applications generally do not use trend information, or what structural geologists refer to as anisotropy, and gradient 13 type information such as horizon dip data, with polarity, or direction, which the structural agents do in this study, however 14 these diverse applications do have some common elements that software agents are well suited to. The problem domains have 15 multi-scalar environments; molecular to planet scale, with local or global model element interactions, and non-linear, multi-16 source physical dependencies. Agents could be interacting at molecular scale with quantum-mechanical, ionic and 17 thermodynamic influences, for example, for protein-folding (Semenchenko et al., 2016; Nelson et al., 2000), for a visual 18 demonstration of molecular agent simulation see: https://www.youtube.com/watch?v=4Z4KwuUfh0A or at galactic scale 19 http://www.gravitysim.net/index.html." 20 21 section 1.3 this is a good description of the problem of interpreting a 3D subsurface geologic structure given sparse data. Consider moving it, or a suitably edited version of it, 22 23 up near the beginning of the paper, where you are framing the fundamental problem to be 24 solved. 25 This was moved, now in the abstract and general introduction section. 26 27 section 1.4: It is great that there are example codes provided. 28 Actually, most of the work over the last few years was in developing these codes. 29 30 section 2, 2-5 This sentence or something like it would be helpful in the first paragraph of 31 the paper to educate the reader on what is meant in this context by 'geological modelling'. 32 33 Yes, agreed. See earlier comments, this was reworded and moved up. 34 35 6, 11-12 and fig 1: nice illustration of the challenge, and how different algorithms can 36 come up with very different solutions given the same data. Suggest moving this figure up 37 closer to the opening of the manuscript. 38 39 Indeed, Figure 1 is now earlier as part of the introduction. 40 41 8, 19 I'm not familiar with the terms 'line and fabric densification' or 'contact estimation' -42 some definitions would help here. 43 44 Section 2.1 Structural Agents, has a more information now and less jargon. Formlines are defined 45 and some jargon deleted. 46 47 (P.10,L.25) "Typically, structural trends are manually traced in 2D, on maps and cross sections, with what are referred to as 'form lines' 48 49 that match the local planar fabric observations." 50 51 section 2.1 This section says a bit about what tasks agents are used to achieve (e.g., interpolation) but so far one thing that is missing is an explanation of why agent-based 52

algorithms might be expected to do a better job than the more conventional alternatives. 2 For example, is the idea that a large number of agents, each implementing a fairly simple 3 set of rules, would be easier to program, or faster to run, or less costly in memory resources, than a running a single but presumably more complicated 'global' algorithm? Or 4 that an agent based approach makes it easier to adapt sampling density according to local 5 features in the data, because agents can 'signal' one another or spawn new agents in 6 7 response to finding something 'interesting' (by whatever criteria) in the data? Whatever 8 the case, this paper would really benefit from a concise statement about how and why agents might be expected to provide a better solution (and along the way, to tell us a bit 9 about what the standard, non-agent algorithms and methods look like, and why they are 10 11 problematic). 12 13 Yes, I agree a better explanation was needed. I added new text going into more detail.... 14 15 (P. 10, L.17-21) "The major benefit of spatial agents is that they can be programed to act as a swarm. That is, they can act 16 collectively, having cohesion with their local neighbours, thus providing the spatial continuity required to construct 17 continuous features. The swarm may also be given shape-based rules, such as, keep members on a local plane or within a specified degree of curvature. This is difficult to achieve with a global algorithm; inverting a matrix containing all
 constraining data and properties."

With the re-organization of the paper and new text in the introduction, and better explanation of the structural
agents the issue of how and why the agents might provide a *'better and more efficient solution'* is partially
addressed. It is more explicitly dealt with in new text at (P.7,L.1-6)

"The ability to operate in a non-centralized control structure, being sensitive to other neighbours conditions and geometric
states as well as their ability to respond to local or globally changing conditions may give spatial agents an advantage. Their
independence allows them to operate as individual elements, for example a single point observation, or to work collectively
as a team or 'swarm'. This allows the application of agent rules that may determine local cohesion levels and shape
characteristics as well as changes of state depending on specific conditions such as moving in a direction, stopping, or
spawning other processes."

32

8, 21-23 The idea that agents construct triangular meshes from point data is alluded to 34 here, and shown in Fig 4, but the description is relegated to an appendix. Yet this actually 35 seems important enough to consider pulling into the main text. At this point in the 36 manuscript, the end of page 8, the text has not yet offered much description of what it is 37 these spatial agents actually do. It would help to have a description of at least one 38 example of a particular agent algorithm, to help readers picture how this is meant to work. For example, does an agent 'move' around the space at random until it finds a 39 40 point, and then create a triangular mesh element? A verbal description of the algorithm, 41 even if it is just one out of several algorithms that are used in the accompanying software, would really help a lot in understanding how this approach works. We learn later, in 42 section 3, about how some of these agents operate, but by giving an example earlier in 43 44 the manuscript you can give the reader a more intuitive sense of what this kind of agentbased 45 approach involves. 46

New explanations to give a simple procedure description of the meshing process as an example of
how agents can work, see (P.11,17 to P.12,L.6). A new pseudo-code is also provided in the
Appendix-B.

- 51 11, 6 what is a 'free' agent and how does it differ from any other type(s) of agent?
- 52

**A free agent is not yet part of a swarm. See section 2.2 Agent communication (p.14,L.5). A better definition was given.**

P.11, 12-13 can you say more about how an agent would perform interpolation? Would it be
limited to a particular neighborhood of points, and if so how is that neighborhood defined?
Again, why would this be preferable to just doing a global IDW or SLERP?

**8 New text added giving more details of how the swarm behaves**

(P.15,L.4-8) "It is in this way that an agent can define a local neighbourhood as a local swarm, not just by proximity, but also with geometric properties such as orientation. An agent might be very close to its neighbour but may not be selected to be in the swarm because it is oriented at too high an angle thus promoting agents that are near co-planar to be working together. Agent interpolation is *not actually replacing more classical schemes*. SABM's are more of a framework in which interpolation and other spatial operators can be called from as needed."

- 14 15 The interpolator (IDW/SLERP) coupled with the cohesion functions produce swarms that try to have 16 their members match data orientation and/or position, but also keep the local swarm looking like a 17 surface, or close to co-planar. This happens with no gridding or mesh. It is however not a 18 completely data driven approach. Individual structural agents may not even see the data, but they may see their neighbours and become part of their swarm, that have some members who see the 19 20 data. Members who see the data can be influenced by it, and then transfer properties to the whole 21 swarm. This is a way of extending the data without producing a drop off weight value with 22 increased distance. The application will need to balance this data weighting versus swarm cohesion 23 effect. Currently the user balances this manually, by setting distance and attraction parameters 24 until a desired form is achieved, so there is a knowledge guided approach acting here. These parameters are described in the code comments. I did not discus this aspect in detail in the paper 25 26 because it opens the door to the knowledge versus data issue. How to decide what the relative 27 influence of knowledge versus data will be. I honestly would rather leave it for the next phase of 28 research. In the end what happens if we make a system that can deliver a nice version of what is 29 in your head while attempting to respect all the data? We want to go beyond this to making all the 30 end member models that at least make geological sense while coming close to respecting all the 31 data. Maybe, only a sub-set of these matches what is in your head. Maybe none of them do, which 32 is good to clear the head of bad ideas. So far, no system dealing with sparse data, that I am aware 33 of, can achieve this. 34
- Section 2.3: now we are finally getting a look at advantages of agent-based methods over
  traditional approaches. It would be really helpful to give a brief summary of this
  information in the introduction, so readers understand the motivation for this alternative
  approach.
- 39

**40 Yes, for sure. This material was put earlier into the introduction.**

13, 6 Would you not have issues with inter-agent communication in a parallel distributed 43 system? It certainly seems worth experimenting on, but I know that the somewhat similar 44 method of discrete-event simulation has challenges with parallel operation because you 45 can't predict in advance when initially independent operations on different parts of the 46 model space will end up triggering simultaneous or conflicting modifications to the same 47 data (still, the comment about 'yet to be tested' is fair enough, and I agree). 48

Yes, also I agree it is not a given. I do think separate swarms could be assigned separate GPU's,
but their members do change often, which has to be updated for that processor. If each search
agent had its own GPU perhaps that would work, but again lots of inter-agent communication is
needed. There are many other optimizations that could be made as well such as not processing agents that have reached a stable configuration or have reached a stop-criteria. Each geometric 2 function could be assigned a processor (CPU/GPU) since most of them act like a black box. 3 Although not implemented herein, ultimately, the structural interrogator agents are supposed to be temporary. They would be there to act as control points for local meshing. Once the mesh is built, 4 they are not needed as all the geometry information (normals; eigenvectors; swarm membership 5 6 history), can be transferred the mesh nodes. Currently the codes *geoswarm* and *mesh* are 7 separate. This is all for potential future research. I have also not discussed implementation of 8 uncertainty handling but that could be a big part of the future developments. In my opinion, it was beyond the scope of this concept paper since uncertainty management would need a new set of 9 properties and functions beyond what the existing geometry engine provides currently. 10 11 12 13 13, 18 This is interesting - can you say more about 'interrogators or observation sets'? Does this mean for example that you use agents both to represent observational features 14 15 (such as a known dip at a particular (x,y,z) location) and to perform actions like 16 interpolation? 17 18 Yes, that is true agents can be data or interrogators (essentially search agents) but not at the same time. This is 19 clarified in 20 (P.17.L.15) "...spatial agents may represent control data, interrogators or estimated solutions. They could also morph from one type to another. For example, a data agent could extend itself by expanding incrementally along 21 the dip plane directions into estimation points. They may have properties for tracking local swarm or global 22 23 states, continuously checking..." Also, this is made clearer with an expanded description of section 2.1 Structural 24 Agents. 25 26 Section 3: it is great that there are codes provided for each of these examples. 27 28 Indeed. I wanted to show it is not just a hunch, but works in small ways. 29 30 15, 2-5 This specific example of an agent's behaviour is really helpful. Consider presenting this or another example much earlier in the manuscript, not as a comprehensive 31 32 description of the different agent types and their behaviours (you already have that material down here in section 3), but just to give the reader a general idea of what you 33 34 mean when you refer to 'spatial agents' in the paper's introduction. (Something like: 'For 35 example, to help interpolate the surface of a dipping geologic unit, one could define an agent that moves randomly through the data volume until it encounters a dip observation, 36 at which point it uses a local interpolation algorithm to spawn new dip markers in the area 37 around the observation'...or something along those lines, though actually Mesh agents 38 39 might be an easier example to understand). 40 41 This was fixed in an expanded step-by-step description of the mesh program (see P.11,L.15 to 42 P.12,L.6). Also, pseudo-code added to Appendix B. 43 44 Technical corrections: 45 2, 7 17 inc fixed (EdK) 46 4, 6 inc fixed (EdK) 47 5, 2 inc fixed (EdK) 48 12, 10 inc (also, not obvious what 'contributions from multi-scalar and deep multiproperty 49 data' means) 50 Line added with examples for clarification ... (P.16,L.7) "Preserving contributions from multi-scalar and deep 51 multi-property data, such as fold shape parameters, or geophysical rock properties." 52

- 1 12, 12 inc fixed (EdK)
- 2 12, 17 inc fixed (EdK)
- 3 13, 7 inc fixed (EdK)
- 4 15, 13 criterion fixed (EdK)
- 5 19, 7 inc fixed (EdK)
- 6 20, 3 inc fixed (EdK)
- 7 20, 9 was fixed (EdK)
- 8 20, 19 inc fixed (EdK)
- 9 21, 3 inc fixed (EdK)
- 10 11

**12 **Comment on gmd-2021-66**

Guillaume Duclaux (Referee)

Referee comment on "Spatial Agents for Geological Surface Modelling" by Eric A. de Kemp,
Geosci. Model Dev. Discuss., https://doi.org/10.5194/gmd-2021-66-RC2, 2021

Review of "Spatial Agents for Geological Surface Modelling", by Eric de Kemp.

This manuscript presents an innovative contribution to the challenging task of generating 20 3D surface model of complex geological terrains. The generation of 3D models (in the 21 sense of 3D maps and not 3D thermo-mechanical models) is of considerable interest to 22 the broad structural geology and tectonics research community studying the geometry of 23 geological units/objects and contacts in deformed regions, and is also of economic 24 significance for the ressource industry. The author presents here a new surface modelling/meshing method based on spatial agents which has the potential to overcome 25 26 some of the limitations inherent to the more classical implicit methods used in 3D 27 geological modelling. The spatial agent methods have been used now for about a decade for modelling features in a wide range of fields outside of Earth sciences, but such 28 29 methods, here involving structural agents, have never been used specifically for resolving 30 complex geological geometries in 3D by satisfying contacts and structural observations. 31 The paper first introduce the agent method and briefly reviews published literature on agent applications in various fields. The second section of this contribution presents the 32 33 challenges faced when applying current geological surface modelling techniques to 34 complex geological structures with sparse control points. Section 3 presents spatial agents example and how they can be used for solving surface modelling problems in structural 35 geology in order to insure that surface topology is sound and verifies fabrics observations. 36 37 The 6 main structural agents programs presented by the author represent building blocks that could be combined to eventually generate complex surface geology models. Section 4 38 39 discuss the future of such methods. This work doesn't claim that spatial agents are the 40 ultimate solution for resolving complex surface models, but it provides solid evidence that 41 used in conjunction with other Loop 3D tools it could definitely improve the surface model 42 building workflow and insure structural observations are respected. 43 44 The limitations of the current implementation are explicitly presented in the conclusions, which 45 keeps this contribution honest.

Thank you for seeing that I don't claim to create a whole new system solution, just enough to get
the conversation going and hopefully get someone interested to work on this more intensely,
perhaps as a graduate or post-doc study.

The manuscript is well written and articulated. It contains 7 figures and 3 appendices.

some figures should be better called in the text. I do not have any major concern with 2 respect to this contribution that seems to represent a solid proof of concept and tile the 3 path for future applications of spatial agents for 3D geological modelling. This work is of broad interest to the community, and especially relevant to the those interested in 3D 4 structural mapping, tectonic interpretation of complex terrains, and the community 5 6 involved in 3D surface modelling in general. As such it seems worth publication in GMD 7 and well suited for this special issue on Loop 3D modelling. I recommend accepting the 8 manuscript with minor revisions and a few technical corrections (see minor comments 9 below). 10 11 Super! Thanks for all your work to check the manuscript. 12 13 I think many of the concerns of Reviewer 2 (Guillaume Duclaux) are dealt with in my response to 14 reviewer 1 so please check my responses there as well. 15 16 Minor comments: 17 18 + NetLogo-3D: this seems to be the correct spelling. It changes throughout the manuscript (starting in the abstract). Could you please insure spelling is correct and 19 20 consistent? (p5, p13, p17, p27, p28) 21 22 Fixed (EdK), and changed to NetLogo 3D throughout. 23 + p2, line 13: there is a typographical error for "conductivity" 24 25 Fixed (EdK). 26 + p4, line 24-25: I would suggest the author edit slightly the last sentence of the page. I 27 believe a model never reconciles all the data... Remembering Box famous aphorism "All 28 models are wrong, but some are useful" we can safely say that no model will reconcile and 29 respect all data. I would possible write down : "[...] explanatory model that aims to 30 reconcile and respect all the available data". 31 Fixed (EdK), re-worded (P.8,L.18) "that aims to reconcile and respect all the available data...". 32 33 I would say that an exact fitting model is exactly matching the data, that could be just 1 data point, that was used 34 to make it BUT that the model is still wrong because where the model is estimating away from data it can have 35 high degree of various errors... spatial, gradient, conceptual ... 36 37 + p5, line 4: section 1.4 title could be revised a bit... in fact it rather presents the outline 38 of the paper. Maybe something like "Outline and demonstration code" ? 39 I re-organized so the paper outline comes at the end of the general introduction and before section 40 1.1. 41 + p7, Figure 1: I would love to see what surface model spatial agents would generate 42 using the data provided in a). Could the structural agents programs presented here 43 resolve this surface in a way that satisfies the control structure in b)? 44 45 I dream about this! So far, I have run several experiments, on that data, that produce local 46 swarms wrapping around the on-contact control points, but they conflict with the stratigraphic 47 levels as there is no multi-level code yet. If I use a single level, it gets close to a reasonable 48 solution, coaxial structures are preserved but the spatial continuity is not very good. I need better 49 cohesion algorithms to glue the agents, or start to mesh them and freeze the local solution before 50 moving on. Basically, it needs the next phase of research to get a good result. It would need a 51 dedicated effort I think, and perhaps a more industrial strength agent environment that can handle

| 1
      | larger coordinates than NetLogo (such as Massive , or Repast Symphony https://repast.github.io/screenshots.html).                                                                                                                                                     |
|----------------------------|-------------------------------------------------------------------------------------------------------------------------------------------------------------------------------------------------------------------------------------------------------------------------------------|
| 5
      | + p8, line 23: the notion of continuity for spatial agents is not very to me, even when looking at Fig 4. I believe the author should explain what is meant in more details.                                                                                                        |
| 7
      | New text was added (P.11,L.17 to P.12, L.6) to better explain the meshing, and continuity clarified with "overall continuity, meaning the surface has no holes or branches."                                                                                                        |
| 10
| + p11, lines 3-4: I reckon an UML diagram or some schematics illustrating agents interaction would be helpful to those like me who are not familiar with such functions.                                                                                                            |
|                            | Yes, agreed, I created a new general agent communication diagram, see Figure 5. The caption and connection labels should made things clearer.                                                                                                                                       |
| 15
   | + p12, lines 11-12: a reference to Figure 1 would be great here.                                                                                                                                                                                                                    |
| 17
   | + p12, line 12-14: I totally agree! Geophysics alone is definitely not designed to assess                                                                                                                                                                                           |
| 20
         | What is good is that we can get gradient information and feed that to the agent system.                                                                                                                                                                                             |
| 22                         | + p14, line 4: please add the missing "." in the caption between "data" and "Depending"                                                                                                                                                                                             |
| 24
         | + p14, line 15: please add some comas : [] the program, its intended behavior, and the main []                                                                                                                                                                                      |
| 26                         | Done (EdK).                                                                                                                                                                                                                                                                         |
| 27
         | + p15, line 2: a reference to Figure 2 should be added here.                                                                                                                                                                                                                        |
| 29                         | + p16, line 2: a reference to Figure 3 should be added here.                                                                                                                                                                                                                        |
| 30
   | + p29, line 21: What is the polarity of the rock unit is unknown? High grade metamorphic                                                                                                                                                                                            |
| 33
   | rocks generally have no evident markers for polarity. Is it set to 0? NaN?
Done, herein we set unknown to 0, but in other applications (e.g. Gocad SPARSE plugin) the local
orientation label is 'overturned' and set to 1 (logical) for upright, 0 for overturned and -1 for |
| 36
         | unknown (EdK).                                                                                                                                                                                                                                                                      |
| 38                         | + references formatting in the text need to be formatted according to the journal                                                                                                                                                                                                   |
| 39
         | Done, many fixes (EdK).                                                                                                                                                                                                                                                             |
| 41
         | Comas are missing between author names or et al. and the year. Some                                                                                                                                                                                                                 |
| 43
| references have typos (i.e. p3 line 22 "Motieyan. and Mesgari", or p13 line 14 "from dekemp").
Done, many fixes (EdK).                                                                                                                                                           |
| 47                         |                                                                                                                                                                                                                                                                                     |
| 48
| Guillaume Duclaux
Nice, 05/08/2021                                                                                                                                                                                                                                               |
| 71                         |                                                                                                                                                                                                                                                                                     |

[revised manuscript text omitted]

---

## Author Response (AR2)

Dear Dr. Poulet,

Thank you for going through the paper for technical and grammatical corrections. I believe all changes have been made. The only issue is for the GSC contribution number

Please see all changes in Red below your comments.

Regards,

Eric de Kemp

Dear Dr de Kemp,

Thank you very much for the extensive modifications of the manuscript (including reorganisation of the abstract, introduction, as well as new figure) which address all comments from the reviewers. As everyone had noted previously, this paper is a valuable contribution and you have now successfully broadened its potential audience with clear explanations for non-experts in the field. I also agree with your choice of leaving some points out of this work, like the "knowledge vs data" issue.

I am therefore pleased to let you know that the paper will be accepted for publication after some technical corrections in the text. Please check your manuscript for typos and grammatical mistakes (e.g. using tools like gammarly.com):
• P.1, L.13: "[have] provided" (subject is plural)
Fixed (EdK).
• P.1, L.15 "[has] been trying" (subject is singular. Expressions like "along with" don't act as coordinating conjunctions so the subject remains "The Geological Survey of Canada" only)
Fixed (EdK).
• P.1, L.16: comma after "however"
Fixed (EdK).

• P.1, L.17: "involve" (subject is plural)
If subject is plural should verb not conjugate to a plural as well "involves"?
• P.1, L.17 "develop[ing] of surface components"
Fixed (EdK) but sounds weird. Feels like "of" should not be there?

• P.1, L.20: rephrase "to develop reasonable starting framework geological models" if possible to explicit the link between "starting", "framework" and "geological models" (better explained in your introduction)

Slight nuance changed to "develop geologically reasonable starting framework models".
• P.2, L.20: "explore[r]s"
Fixed (EdK).
• P.3, L.9: "bias for", check use of transitive verb "to bias" (you might be right, but given the typos above I'm pointing this one for checking, being myself a non-native English speaker...)
Changed to "...tend to be biased towards these...".
• P.8, L.8: "the focus [is]"
Yes, corrected ... "The focus is on visualizing and modelling...".
• Acknowledgements: "NRCan Contribution Number [xxx]"
May need to be updated at galley proof stage (EdK).

Best regards,

Thomas Poulet.